# KAT8-mediated H4K16ac is essential for sustaining trophoblast self-renewal and proliferation via regulating CDX2

Shilei Bi [1,2,3,4,12], Lijun Huang[1,2,3,4,12], Yongjie Chen[5,12], Zhenhua Hu[1,2,3,4], Shanze Li[6,7], Yifan Wang[1,2,3,4], Baoying Huang [1,2,3,4], Lizi Zhang [1,2,3,4], Yuanyuan Huang[1,2,3,4], Beibei Dai[1,2,3,4], Lili Du [1,2,3,4], Zhaowei Tu[1,2,3,4], Yijing Wang[6,7], Dan Xu[6,7], Xiaotong Xu[6,7], Wen Sun[1,2,3,4], Julia Kzhyshkowska[8,9,10], Haibin Wang [11] ✉, Dunjin Chen [1,2,3,4] ✉, Fengchao Wang [6,7] ✉ & Shuang Zhang [1,2,3,4] ✉

Abnormal trophoblast self-renewal and differentiation during early gestation is the major cause of miscarriage, yet the underlying regulatory mechanisms remain elusive. Here, we show that trophoblast specific deletion of *Kat8*, a MYST family histone acetyltransferase, leads to extraembryonic ectoderm abnormalities and embryonic lethality. Employing RNA-seq and CUT&Tag analyses on trophoblast stem cells (TSCs), we further discover that KAT8 regulates the transcriptional activation of the trophoblast stemness marker, CDX2, via acetylating H4K16. Remarkably, CDX2 overexpression partially rescues the defects arising from *Kat8* knockout. Moreover, increasing H4K16ac via using deacetylase SIRT1 inhibitor, EX527, restores CDX2 levels and promoted placental development. Clinical analysis shows reduced KAT8, CDX2 and H4K16ac expression are associated with recurrent pregnancy loss (RPL). Trophoblast organoids derived from these patients exhibit impaired TSC self-renewal and growth, which are significantly ameliorated with EX527 treatment. These findings suggest the therapeutic potential of targeting the KAT8-H4K16ac-CDX2 axis for mitigating RPL, shedding light on early gestational abnormalities.

The trophoblast from trophectoderm of blastocyst is the epithelial cells in placenta and plays essential roles in oxygen and nutrient transport, hormone secretion and immune defense that required to maintain fetal growth during pregnancy[1]. Trophectoderm, the outer layer of the blastocyst, harbor trophoblast stem cells which differentiates the precursor of all trophoblast cell types and transit into cytotrophoblast (CTB) and syncytiotrophoblast (STB) during implantation[2]. Disruption in trophoblast stemness maintains and differentiation in the early stages of gestation are recognized as the root cause of various complications associated with pregnancy, including

miscarriage, preeclampsia, and intrauterine growth restriction. Notably, a comprehensive exploration of trophoblast defects during the early gestational stage, particularly peri-implantation period, remains limited compared to later gestational phases[3–5]. Consequently, investigating the underlying regulatory mechanisms holds the potential to unveil the origins of placental malformations and may provide a promising target for improvement.

The study of early human placental development faces practical and ethical challenges, prompting the use of animal models and in vitro systems, including trophoblast stem cell and organoid models, to

A full list of affiliations appears at the end of the paper. ✉e-mail: haibin.wang@vip.163.com; gzdrchen@gzhmu.edu.cn; wangfengchao@nibs.ac.cn; shuang1zhang@gzhmu.edu.cn

investigate trophoblast lineage differentiation and function[6–8]. Mice have served as the primary models for examining placental development. In mice, the polar trophectoderm (TE) covering the inner cell mass (ICM) continues to proliferate, forming the extra-embryonic ectoderm (ExE) containing trophoblast stem cells and the diploid ectoplacental cone (EPC)[9]. Concurrently, mural cells cease dividing and transform into trophoblast giant cells (TGCs). Subsequent differentiation gives rise to various trophoblast cell types within the mature chorioallantoic placenta, including the labyrinth, spongiotrophoblast, and glycogen cells[9]. Through screening over hundreds of mouse knockout (KO) lines, Hemberger group revealed that the majority of the lines with embryonic lethal phenotype exhibited placental abnormalities, highlighting the critical role of placental defects in contributing to abnormal embryo development[10]. Despite the identification of certain regulators of trophoblast lineage specification, including *Cdx2, Esrrb, Eomes, Gata2/3*, and *Tead4*[11–14], the precise mechanisms governing TE progenitor self-renewal and ExE development during early placentation remain largely enigmatic.

KAT8 (Lysine acetyltransferase 8), also known as MOF (Males Absent of the First) or MYST1, is a member of the MYST family of histone acetyltransferases (HATs), and it plays a pivotal role in acetylating H4K16[15].H4K16ac is a highly abundant activating modification and play important roles in modulating chromatin structure and transcriptional activation[16,17]. The regulation of KAT8-H4K16ac axis is finely tuned, with subtle changes in H4K16ac levels yield distinct and readily observable phenotypic effects[18]. KAT8 and H4K16 acetylation have been implicated as essential regulators of various cellular processes, encompassing DNA damage response[19], cell death, apoptosis[20–22], and autophagy[20,22–25]. Extensive research has elucidated the pivotal role of H4K16ac in maintaining the pluripotency of embryonic stem cells (ESCs)[16,26]. Notably, in mice, the absence of KAT8 results in a widespread depletion of H4K16 acetylation, leading to embryo lethality at implantation[27]. Additionally, KAT8 and H4K16ac have emerged as novel markers for active enhancers and promoters in genes crucial for preserving ESC identity[16,20,28]. However, their potential roles in placenta development, particularly in the regulation of the trophoblast lineage, remain unexplored.

In this work, we investigate the role of KAT8-H4K16ac in trophoblast stemness maintenance, proliferation and differentiation while unraveling the underlying regulatory mechanisms. We also explore the clinical relevance of KAT8-H4K16ac and its potential as a therapeutic strategy for mitigating placenta-derived pregnancy loss.

## Results

### *Kat8* knockout in trophoblast leads to abnormal placentation

To systematically elucidate the roles of KAT8-H4K16ac in early placental development, we collected mouse embryo at peri-implantation stage, ranging from embryonic day (E) 4.5 – E9.5, and examined the spatiotemporal expression patterns of KAT8 and H4K16ac during mouse placental development. Immunohistochemistry (IHC) staining showed a widespread expression of KAT8 and H4K16ac in various tissue structures, including TE, ExE, EPC, chorion, and chorionic plate (CP), at different stages of placental development (Fig. 1a).

To investigate the function of KAT8 in placentation, we generated *Kat8^{f/f}* transgenic mouse model by inserting Loxp fragments into exons 2 and 3 of the *Kat8* gene (Fig. 1b; Supplemental Fig. 1a). Subsequently, we crossed *Kat8^{f/f}* with *Elf5-Cre* mice, which specifically function in trophoblasts[29], to achieve trophoblast-specific deletion of *Kat8* (*Kat8^{d/d}*) (Fig. 1c). According to Mendelian inheritance, this breeding strategy would result in approximately a 1/4 probability of obtaining *Kat8^{d/d}* offspring. However, upon genotyping the offspring, we failed to get litters with *Kat8^{d/d}* genotype, suggesting that the *Kat8^{d/d}* in trophoblast cells leads to embryonic lethality (Fig. 1d; Supplemental Fig. 1b). To identify the specific stage at which the pregnancy defects occurred, we examined the pregnancy status at different days of early gestation. In

the *Kat8^{f/f}* female mice crossed with *Kat8^{f/+};Elf5-Cre* male mice, the pregnant uteri appeared normal on E6.5, E7.5, and E8.5, as indicated by the gross number of the implantation sites, respectively (Fig. 1e). However, absorbed implantation sites were clearly observed after 10 days of pregnancy, indicating the placenta defects occurred earlier than this stage (Fig. 1e). We then carefully dissected and stained the embryos from *Kat8^{f/f}* pregnant mice crossed with *Kat8^{f/+};Elf5-Cre* male mice, spanning the developmental stages from E5.5 to E8.5. At E8.5, we observed that the chorion of *Kat8^{f/f}* embryos fused EPC to form chorionic plate, while the development of *Kat8^{d/d}* embryos appeared completely disorganized, lacking placenta morphology (Fig. 1f; Supplemental Fig. 1c). At E7.5, *Kat8^{f/f}* embryos exhibited the development of structures such as the chorion and EPC, which are precursors of future placental structures (Fig. 1f; Supplemental Fig. 1c). In contrast, *Kat8^{d/d}* embryos did not show the presence of these structures, and this lack of ExE and EPC in *Kat8^{d/d}* embryos was observed as early as E5.5, indicating the critical role of KAT8 in placenta development (Fig. 1f; Supplemental Fig. 1c).

To further assess the impaired trophoblast development in Kat8^{d/d} embryos, we examined *Kat8^{f/f}* and *Kat8^{d/d}* embryos at E4.5 and E5.5, prior to the disappearance of trophoblasts. Immunofluorescent staining analyses revealed a substantial depletion of H4K16ac in *Kat8^{d/d}* embryos at E4.5 (Fig. 1g). Furthermore, the expression of Ki67, a marker of cellular proliferation, was undetectable in a subset of H4K16ac negative cells of *Kat8^{d/d}* ExE at E5.5 (Fig. 1g), indicating a decline in the proliferation of trophoblast cells in *Kat8^{d/d}* embryos. These data demonstrate that the embryonic development failure observed in *Kat8^{d/d}* mice is attributed to defects in trophoblast lineage, indicating the significance of KAT8 in early placenta development.

### Impaired trophoblast stemness upon *Kat8* deletion

To investigate the role of KAT8 in the regulation of trophoblast development, we established mTSCs from *Kat8^{f/f}* blastocyst. Subsequently, a Doxycycline (Dox) inducible Cre-expressing vector was introduced exogenously to facilitate *Kat8* deletion (*Kat8^{d/d}* hereafter) (Fig. 2a). In this model, floxed *Kat8* alleles can be deleted upon Dox induced expression of Cre recombinase. RT-qPCR, Western blot and immunofluorescent staining revealed efficient *Kat8* deletion and significant reduction in H4K16ac expression upon *Cre* induction (Supplementary Fig. 2a, Fig. 2b–d). *Kat8^{d/d}* mTSCs became flattened with enlarged cell size and lost their typical clonal appearance (Fig. 2c). Consistent with the in vivo results, both cell number and the expression of proliferation markers, such as Ki67 and pH3, exhibited a substantial decrease in *Kat8^{d/d}* TSCs (Supplementary Fig. 2b–d; Fig. 2c, d), indicating a significant reduction in cell proliferation capacity.

To gain insights into the gene expression changes associated with KAT8 deficiency and its impact on trophoblast development, we conducted a bulk RNA-seq analysis, comparing the transcriptomes of *Kat8^{f/f}* with *Kat8^{d/d}* mTSCs. Significant alterations in gene expression were observed upon *Kat8* deletion, with 1192 genes upregulated and 1573 genes downregulated ($P < 0.05$, $Log_2FC \geq 1$) compared to genes in *Kat8^{f/f}* TSCs (Fig. 2e). Gene Ontology (GO) enrichment analysis revealed that enriched terms on downregulated genes including DNA replication/recombination/repair, these terms align with the well-established roles of KAT8[19,30], and was validated in *Kat8^{d/d}* mTSCs, which showed increased γ-H2A.X (Supplementary Fig. 2e, f). Whereas the upregulated genes following *Kat8* deletion were predominantly enriched in pathways such as the p53 signaling pathway, focal adhesion, and cellular senescence, among others (Supplementary Fig. 2g, h).

Notably, placenta development was identified as a major enrichment term defined by trophoblast stemness marker genes, such as *Cdx2, Eomes*, and *Ly6a* (Fig. 2f–h). Consistently, Gene Set Enrichment Analysis (GSEA) on the downregulated genes also showed statistically significant differences in gene sets associated with trophoblast

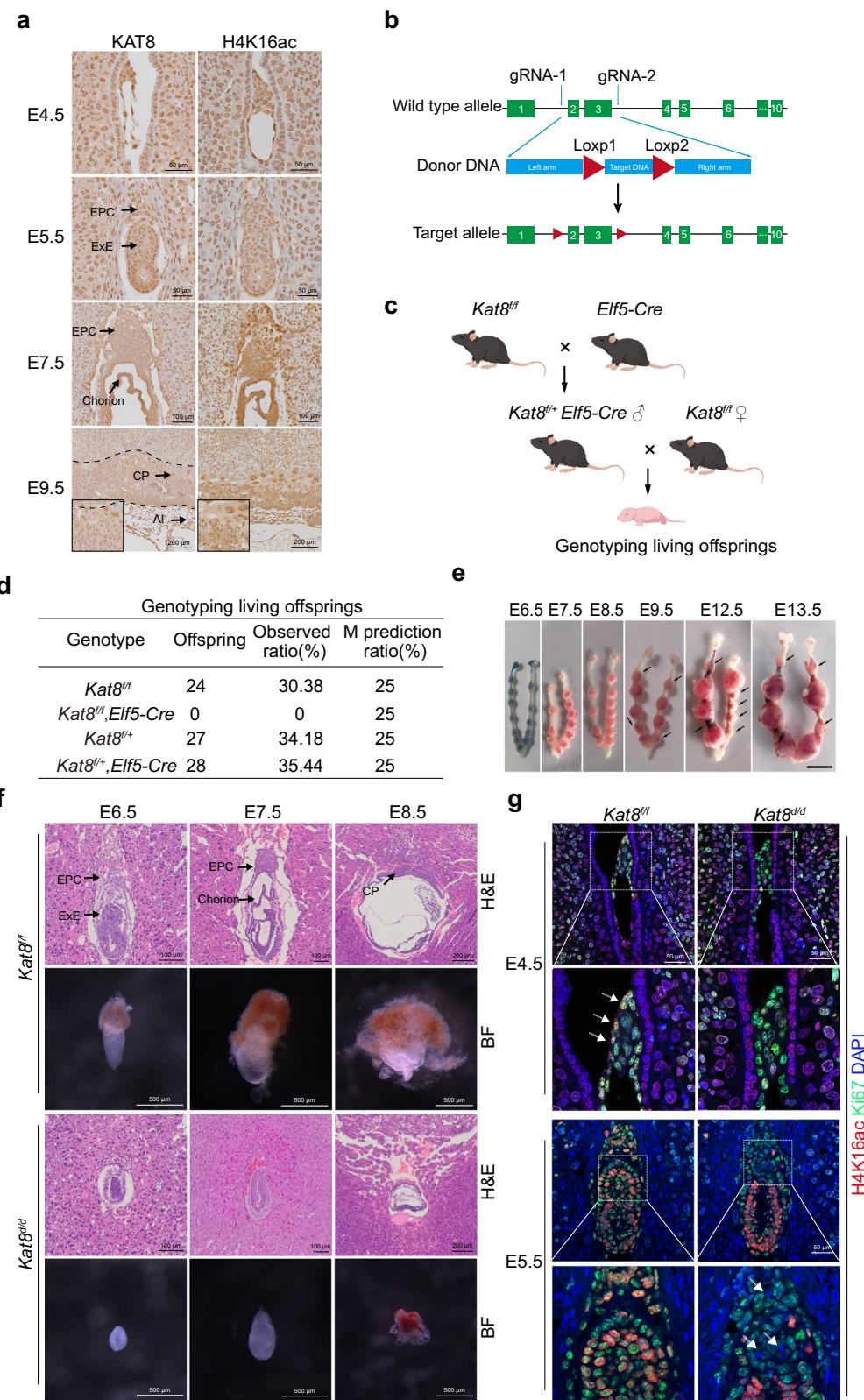

**Fig. 1 | Specific knockout of KAT8 in trophoblast cells leads to abnormal placental development. a** Immunohistochemical staining for KAT8 and H4K16ac at indicated embryonic days during normal placental development. E0.5 was determined as the point at which a vaginal plug was identified. **b** Schematic of generating *Kat8^(f/f)* transgenic mouse. **c** Schematic for generating mice with trophoblast-specific deletion of *Kat8* (*Kat8^(d/d)*). The graphic elements were created by figdraw. **d** Genotype analysis of the offspring from female *Kat8^(f/f)* mice crossed with male

*Kat8^(f/+);Elf5-Cre* mice. **e** Implantation sites in female *Kat8^(f/f)* mice crossed with male *Kat8^(f/+) Elf5-Cre* mice at E6.5-E13.5. Scale bar: 5 mm. **f** H&E staining and brightfield pictures of implantation sites of *Kat8^(f/f)* and *Kat8^(d/d)* mice at E5.5–E8.5. **g** Immunofluorescent staining of H4K16ac and Ki67 in *Kat8^(f/f)* and *Kat8^(d/d)* embryos at E4.5-E5.5. Arrowheads pointing the missed H4K16ac and Ki67 staining in *Kat8^(d/d)* ExE cells. ExE, extraembryonic ectoderm; EPC, ectoplacental cone; CP, chorionic plate. AI, Allantois.

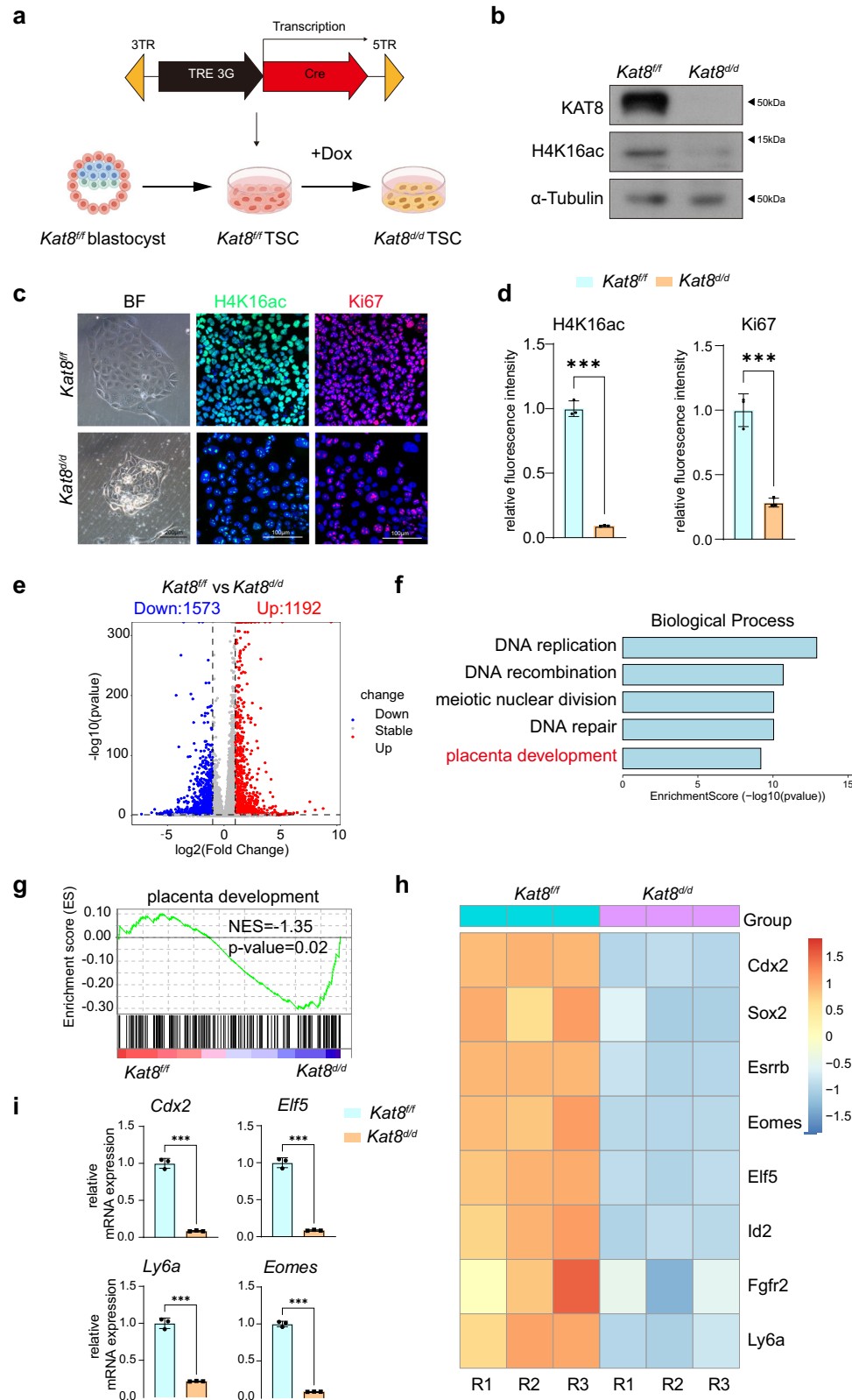

development (Fig. 2g). The decreased expression of *Cdx2, Eomes, Elf5* and *Ly6a* was further validated by qRT-PCR analysis (Fig. 2i). Besides, we also identified a subset of regulators involved in trophoblast differentiation among significantly downregulated genes, including *Gcm1, Syna, Prl3d1, Prl3d2, Prl3d3,* and *Tpbpa* (Supplementary Fig. 2i). These findings suggest a crucial role of KAT8 in regulating the trophoblast development.

## KAT8 catalytic activity essential for CDX2 expression

Given the critical function of CDX2 and EOMES in regulating trophoblast lineage fate determination and differentiation[31], we examined their expression in embryo implantation site of *Kat8^f/f* and *Kat8^d/d* at E5.5 and E6.5, the IF staining showed an apparently decreased expression of CDX2 and EOMES in ExE cells (Fig. 3a). The effect of *Kat8* deletion on CDX2 and EOMES were further demonstrated on mTSCs,

**Fig. 2 | Impaired trophoblast stemness and differentiation upon *Kat8* deletion.**
**a** Schematic showing generation of *Kat8*^d/d^ mTSCs. The graphic elements were created by figdraw. **b** Immunoblot analysis of KAT8 and H4K16ac in mTSCs treated with or without Dox for 48 h (*Kat8*^d/d^ and *Kat8*^f/f^). **c** Representative immuno-fluorescent staining and brightfield pictures of H4K16ac and Ki67 in *Kat8*^d/d^ and *Kat8*^f/f^ mTSCs. **d** The quantitative results of C. Data are representative of three independent experiments (*n* = 3) and the values are normalized to *Kat8*^f/f^ control group. Two-tailed unpaired Student's *t*-test. Error bars, mean ± SEM. H4K16ac, *P* < 0.0001. Ki67, *P* < 0.0001. Source data are provided as a Source Data file. **e** Volcano plot of differentially expressed genes (*P* < 0.05, Log₂FC ≥ 1) in *Kat8*^f/f^ and *Kat8*^d/d^ mTSCs as determined by exact negative binomial test in edgeR package of R.

**f** Gene Ontology analysis for Top 150 differentially downregulated genes by DAVID by Kappa Statistics (*p* < 0.05). **g** GSEA plot showing the enrichment of placenta development related genes in *Kat8*^d/d^ compared with *Kat8*^f/f^ mTSCs. NES normalized enrichment score. *P* = 0.02. **h** Heatmap of mTSCs stemness associated genes between *Kat8*^f/f^ and *Kat8*^d/d^ mTSCs. **i** Quantitative real-time PCR analysis of *Cdx2*, *Elf5*, *Ly6a*, and *Eomes* mRNA levels in *Kat8*^f/f^ and *Kat8*^d/d^ mTSCs. Data are representative of three independent experiments (*n* = 3) and the values are normalized to ACTB. Two-tailed unpaired Student's *t*-test. Error bars, mean ± SEM. *Cdx2*, *P* < 0.0001. *Elf5*, *P* < 0.0001. *Ly6a*, *P* < 0.0001. *Eomes*, *P* < 0.0001. Source data are provided as a Source Data file.

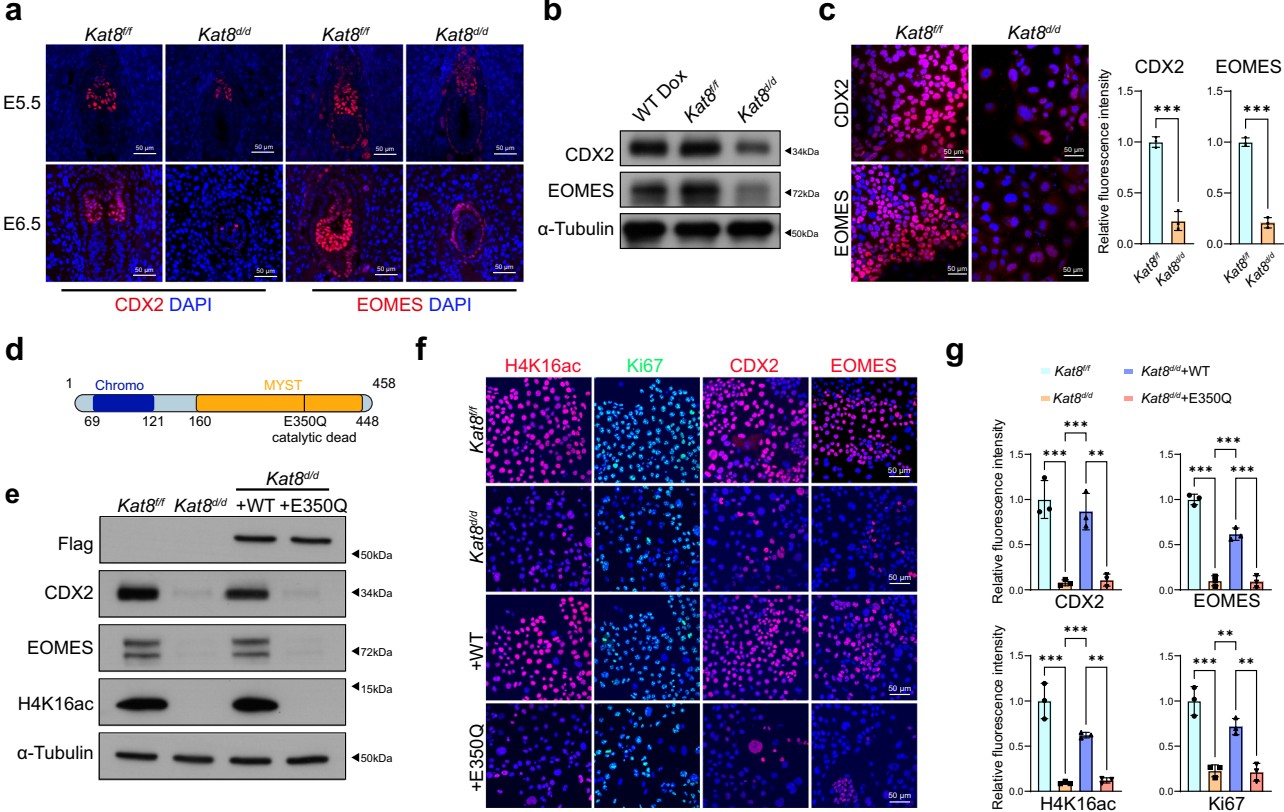

**Fig. 3 | KAT8 is required for CDX2 expression. a** Representative immuno-fluorescence staining of CDX2 and EOMES in *Kat8*^f/f^ and *Kat8*^d/d^ embryos at E5.5 and E6.5. **b** Immunoblot analysis of CDX2 and EOMES in *Kat8*^f/f^ and *Kat8*^d/d^ mTSCs. **c** Left panel: Immunofluorescent staining of CDX2 and EOMES in *Kat8*^f/f^ and *Kat8*^d/d^ mTSCs. Right panel: the relative fluorescence intensity of CDX2+ or EOMES+ signals in *Kat8*^d/d^ compared to *Kat8*^f/f^ mTSCs. Data are representative of three independent experiments (*n* = 3) and the values are normalized to *Kat8*^f/f^ control group. Two-tailed unpaired Student's *t*-test. Error bars, mean ± SEM. *P* = 0.0002 (CDX2, between *Kat8*^f/f^ and *Kat8*^d/d^), *P* < 0.0001 (EOMES, between *Kat8*^f/f^ and *Kat8*^d/d^). Source data are provided as a Source Data file. **d** Schematic representation of the

protein structure of wild-type KAT8 or catalytic dead (E350Q) KAT8. **e** Immunoblot analysis of Flag, CDX2, EOMES and H4K16ac in *Kat8*^f/f^, *Kat8*^d/d^ and *Kat8*^d/d^ transduced with 3xFLAG-WTKAT8 or 3xFLAG-E350QKAT8. **f** Immunofluorescent staining of CDX2, EOMES, Ki67 and H4K16ac in *Kat8*^f/f^, *Kat8*^d/d^ and *Kat8*^d/d^ transduced with 3xFLAG-WTKAT8 or 3xFLAG-E350QKAT8. **g** The quantitative results of F. Data in C and G are representative of at least three independent experiments (*n* = 3) and values are normalized to *Kat8*^f/f^ group and expressed in mean ± SEM. Two-tailed unpaired Student's *t*-test, **P* < 0.01, ***P* < 0.001. Source data are provided as a Source Data file.

with their expression significantly decreased upon *Kat8* depletion (Fig. 3b, c).

To assess whether the crucial role of regulating mTSCs pluripotency depends on the catalytic activity of KAT8, we conducted a knockout of *Kat8* and subsequently expressed either wild-type KAT8 or a catalytically inactive mutant (E350Q) of KAT8 in mTSCs (Fig. 3d). The results showed that decreased expression of CDX2, EOMES, and H4K16ac, as well as Ki67 in *Kat8*^d/d^ mTSCs, were apparently rescued by wild type KAT8, but not E350Q-KAT8 incorporation (Fig. 3e–g), indicating that the catalytic activity of KAT8 is essential for regulating stemness in mTSCs.

## CDX2 is a target gene of KAT8-mediated H4K16ac

As we have confirmed H4K16ac as a critical substrate of KAT8 in mTSCs, we proceeded to perform KAT8 and H4K16ac CUT&Tag experiments to identify downstream target genes that play functional roles in trophoblast development. Consistent with previous publications[32], we found KAT8 and H4K16ac showed a remarkable similarity in the genomic distribution, with a predominant enrichment observed within the promoter regions of genes (Supplemental Fig. 3a, b). A total of 25,944 peaks were identified for KAT8 binding, while 21,324 peaks were associated with H4K16ac. Notably, there were 15,096 overlapping peaks between the two, predominantly distributed

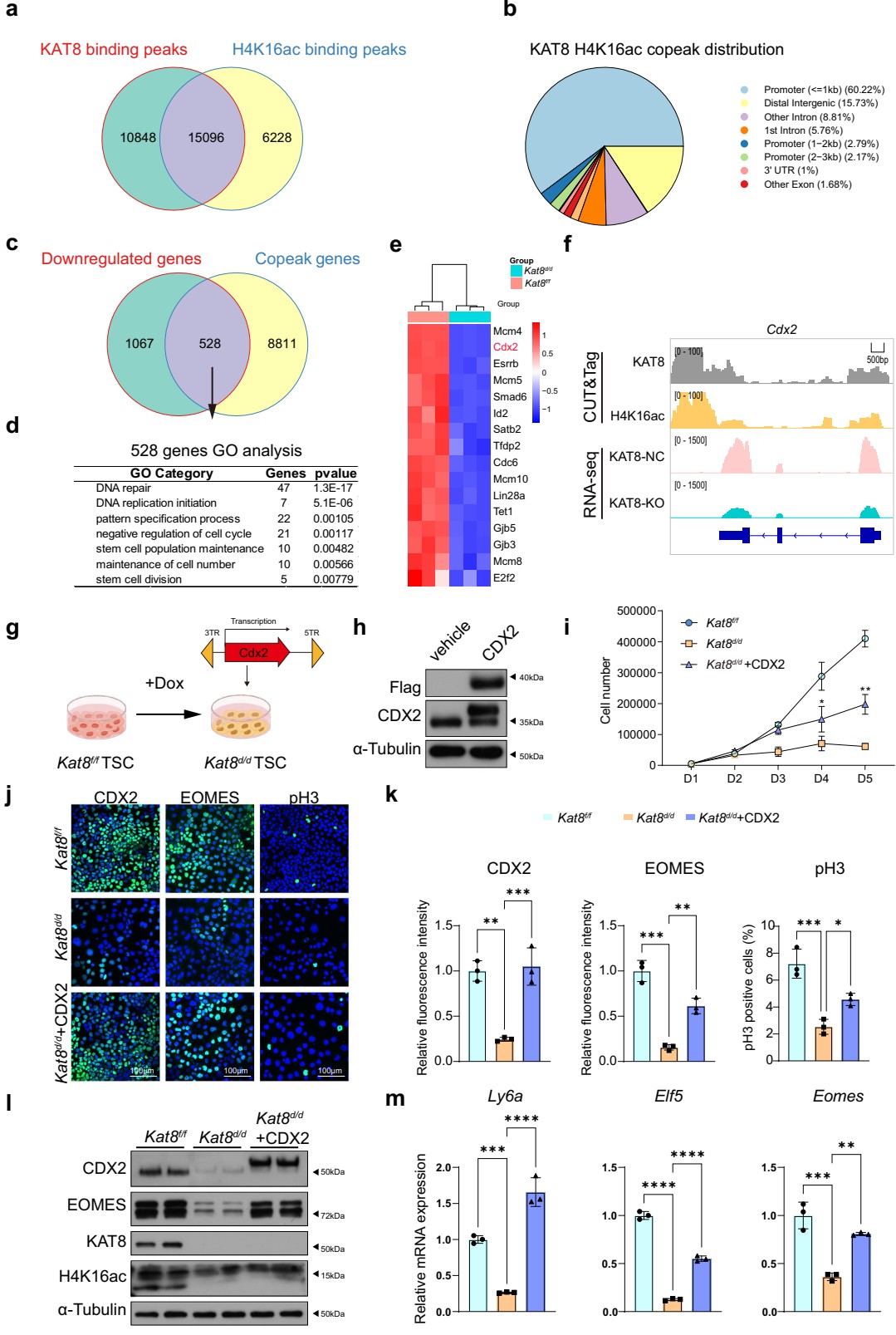

in the promoter regions of the genome (Fig. 4a, b). These findings strongly indicate that KAT8 primarily regulates downstream gene transcription through modulation of H4K16ac.

By integrating KAT8-H4K16ac co-peaks binding genes and downregulated genes of RNA-seq analyses, we identified 528 potential target genes that might be directly regulated by KAT8 through H4K16ac (Fig. 4c). Through GO analysis of these 528 genes, we found

the terms associated with DNA damage repair, DNA replication, and stem cell regulation were enriched (Fig. 4d). Specifically, mTSCs stemness regulators, including *Cdx2, Esrrb, Id2, Lin28a*, and *Tet1*, were identified, while cell proliferation-related genes such as *Mcm4, Mcm5, Cdc6, E2f2*, and *Mcm8* were identified (Supplemental Fig. 3c, Fig. 4e). Further analysis showed that KAT8 and H4K16ac signals are enriched across the promoter region of *Cdx2* (Fig. 4f), suggesting that KAT8 may

**Fig. 4 | CDX2 is a target gene of KAT8-mediated H4K16ac. a** Venn diagram showing the overlap between KAT8 binding peaks ($n = 25,944$) and H4K16ac binding peaks ($n = 21,324$). **b** Genomic distribution of KAT8 and H4K16ac copeaks in mTSCs. **c** Venn diagram showing the overlap between genes with KAT8 and H4K16ac copeaks ($n = 9339$) and genes downregulated upon KAT8 loss ($n = 1595$). **d** Gene Ontology analysis for KAT8 and H4K16ac target genes ($n = 528$) by DAVID by Kappa Statistics ($p < 0.05$). **e** Heatmap of genes associated with stemness and proliferation that are the target genes of KAT8 and H4K16ac. **f** Genome browser view of CUT&Tag-seq signals and RNA-seq tracks for KAT8 and H4K16ac target gene *Cdx2* in mTSCs. **g** Schematic of overexpressing CDX2 in *Kat8*$^{d/d}$ mTSCs. The graphic elements were created by figdraw. **h** Immunoblot analysis of Flag and CDX2 in *Kat8*$^{f/f}$ and *Kat8*$^{f/f}$ transduced with 3xFLAG-CDX2 mTSCs. **i** Cell counts of *Kat8*$^{f/f}$, *Kat8*$^{d/d}$ and *Kat8*$^{d/d}$ + CDX2 mTSCs collected on indicated days. Data are representative of three independent experiments ($n = 3$). Two-tailed unpaired Student's *t*-test. Error bars, mean ± SEM. $P = 0.0463$ (for D4, between *Kat8*$^{d/d}$ and *Kat8*$^{d/d}$ + CDX2), $P = 0.0021$ (for D5, between *Kat8*$^{d/d}$ and *Kat8*$^{d/d}$ + CDX2). Source data are provided as a Source Data file. **j** Immunofluorescent staining of CDX2, EOMES, pH3

in *Kat8*$^{f/f}$, *Kat8*$^{d/d}$ and *Kat8*$^{d/d}$ + CDX2 mTSCs. **k** Right panel: the relative fluorescence intensity of CDX2+ or EOMES+ signals in indicated groups, Data are representative of three independent experiments ($n = 3$) and the values are normalized to *Kat8*$^{f/f}$ control group. Two-tailed unpaired Student's *t*-test. Error bars, mean ± SEM. $P = 0.003$ (between *Kat8*$^{f/f}$ and *Kat8*$^{d/d}$), $P = 0.0026$ (between *Kat8*$^{d/d}$ and *Kat8*$^{d/d}$ + CDX2). Left panel: the percentage of pH3$^+$ cell numbers in mTSCs of indicated genotypes; average cell numbers from five 20X fields were determined. Data are representative of three independent experiments ($n = 3$) and the values are normalized to *Kat8*$^{f/f}$ control group. Two-tailed unpaired Student's *t*-test. Error bars, mean ± SEM. $P = 0.009$ (between *Kat8*$^{f/f}$ and *Kat8*$^{d/d}$), $P = 0.0176$ (between *Kat8*$^{d/d}$ and *Kat8*$^{d/d}$ + CDX2). Source data are provided as a Source Data file. **l** Immunoblot analysis of CDX2, EOMES, KAT8 and H4K16ac in *Kat8*$^{f/f}$, *Kat8*$^{d/d}$ and *Kat8*$^{d/d}$ + CDX2 mTSCs. **m** Quantitative real-time PCR analysis of *Elf5*, *Ly6a*, and *Eomes* mRNA levels in *Kat8*$^{f/f}$, *Kat8*$^{d/d}$ and *Kat8*$^{d/d}$ + CDX2 mTSCs. Data are representative of three independent experiments ($n = 3$) and the values are normalized to ACTB and indicated as the mean ± SEM. Two-tailed unpaired Student's *t*-test, *$P < 0.05$, **$P < 0.01$, ***$P < 0.001$. Source data are provided as a Source Data file.

---

drive the expression of *Cdx2* through H4K16 acetylation, thereby regulating trophoblast cell self-renewal and promoting trophoblast lineage development.

To determine whether CDX2 compensation could restore the trophoblast defects caused by *Kat8* deletion, we overexpressed CDX2 in *Kat8*$^{d/d}$ mTSCs by incorporating flag-CDX2 vector (Fig. 4g, h). A significant recovery in cell growth was observed in *Kat8*$^{d/d}$ mTSCs with flag-CDX2 compared to the *Kat8*$^{d/d}$ mTSCs (Fig. 4i–k). Molecularly, overexpression of CDX2 significantly increases the expression of EOMES and other stemness markers, such as ELF5 and LY6A, in *Kat8*$^{d/d}$ mTSCs (Fig. 4j–m). These findings support the pivotal role of the KAT8-H4K16ac axis in the transcriptional regulation of CDX2, which in turn influences the proliferation, and stemness of mTSCs.

## Elevating H4K16 acetylation with EX527 partially rescues placental defects induced by KAT8 depletion

Previous studies have reported that specific knockout of *Sirt1*, a NAD + -dependent histone deacetylase, in the placenta enhances mTSCs stemness and prevents differentiation[32–34]. Furthermore, the use of EX527 as a SIRT1 inhibitor has been shown to elevate acetylation levels of H4K16[32,34] (Fig. 5a). To evaluate whether increase H4K16 acetylation could restore CDX2 expression and improve embryo defects caused by *Kat8* deletion, we first treated *Kat8*$^{d/d}$ mTSCs with EX527. An evident elevation in H4K16ac and CDX2 levels was noted via western blot and IF upon treatment with EX527 at 20 μM or higher (Fig. 5b–d), suggesting that EX527 restores the expression of CDX2 via increasing H4K16ac. To further validate this finding in vivo, we treated *Kat8*$^{d/d}$ embryos with EX527 and examined the development of trophoblast cell lineages. We initially determined the EX527 dosage that does not induce abnormal embryonic development using wild-type mice. From E2.5 to E8.5, different doses of EX527, ranging from 10 mg/kg to 20 mg/kg were administered daily via intraperitoneal injection to wild-type mice. Upon thorough examination of embryonic development, we found that normal embryonic development persisted without any apparent abnormalities with EX527 dosage as high as 20 mg/kg. Furthermore, administration of EX527 did not impact the expression levels of CDX2 and H4K16ac in the embryos (Supplementary Fig. 4a), reinforcing the safety of EX527. Thus, we commenced daily intraperitoneal injections of 20 mg/kg EX527 on pregnant mice with the *Kat8*$^{f/f}$ genotype, crossed with *Kat8*$^{f/+}$;*Elf5-Cre* male mice, starting from E2.5 onwards (Fig. 5e). Implantation sites were collected at E6.5, E7.5, and E8.5, the IF staining of EPC marker AP2γ, ExE marker CDX2, and the epithelial cell marker CK8 were performed to assess the development and differentiation of trophoblast cell lineages. In line with our previous results (Fig. 1f), we observed the absence of CDX2-positive trophoblast cells in *Kat8*$^{d/d}$ embryos at E6.5 and E7.5. In

contrast, *Kat8*$^{d/d}$ embryos (genotype was confirmed by PCR) (Supplementary Fig. 4b–d) treated with EX527 at E6.5 and E7.5 showed a significant increase in the number of CDX2-positive trophoblast stem cells and AP2γ-positive trophoblast progenitor cells compared to the *Kat8*$^{d/d}$ group (Fig. 5f). Statistical analysis revealed a significant difference in the number of CDX2-positive cells between the EX527-treated group and the *Kat8*$^{d/d}$ group. At E8.5, *Kat8*$^{d/d}$ embryos had completely lost their normal morphology, with only partially absorbed tissue remaining (Fig. 5f, g). By staining for the epithelial cell marker CK8 and the EPC marker AP2γ, we found that EX527 treatment allowed further embryonic development and the differentiation of TGCs in the placenta at E8.5 (Fig. 5f, g, Supplementary Fig. 4e, f). These results indicate that EX527 can partially restore trophoblast cell proliferation and trophoblast stem cell loss caused by *Kat8* knockout. However, the presence of absorbed implantation sites at mid-gestation (E12.5) were observed and no viable *Kat8*$^{d/d}$ embryos were detected by genotyping in EX527-treated mice (Supplementary Fig. 4g, h), indicating that EX527 was unable to rescue the fate of embryonic lethality caused by *Kat8*$^{d/d}$.

## KAT8-H4K16ac-CDX2 axis is conserved in human trophoblast cell

Due to ethical limitations, the previously hTSC lines typically established from placenta villi later than 6 weeks, when CDX2 expression is rarely detected[7,35,36]. To validate whether KAT8-H4K16ac-CDX2 axis findings in murine models apply to human trophoblast stemness regulation, we induced a CDX2-high trophoblast cell line (iTSC hereafter) from H9-hESCs following a previously reported method[37]. As illustrated in Fig. 6a, hESC differentiates into CTBs upon induction with a combination of factors including CYM5541 (the S1PR3 agonist), BMP4 and SB431542 (the selective TGF-β1 Receptor ALK5 Inhibitor)[37]. The application of TM4 medium, including the CYM5541, CHIR99021 (the GSK3β inhibitor), A83-01 (the TGFβ inhibitor), and FGF10, could maintain the stemness status of iTSC[37]. CDX2 expression was confirmed in induced TSCs along with other TSC markers, TEAD4 and TFAP2C, indicating successful establishment of a CDX2-high TSC model (Fig. 6b, c). Using this cell line, we knocked down KAT8 expression via lentivirus-mediated shKAT8 infection. KAT8 knockdown notably decreased H4K16ac and CDX2 expression, as observed through western blot and IF staining (Fig. 6d, e). Additionally, a decrease in Ki67 expression was observed, correlating with slower growth in shKAT8 cells compared to shNC cells (Fig. 6e, f). Overexpression of hCDX2 effectively rescued the impaired cell proliferation caused by KAT8 knockdown (Fig. 6h–j), underscoring CDX2 as a target of KAT8-H4K16ac in human TSC. In line with findings in mTSCs, EX527 treatment restores the expression of both H4K16ac and CDX2 in KAT8 knockdown cells (Fig. 6k, l). These experiments confirm the

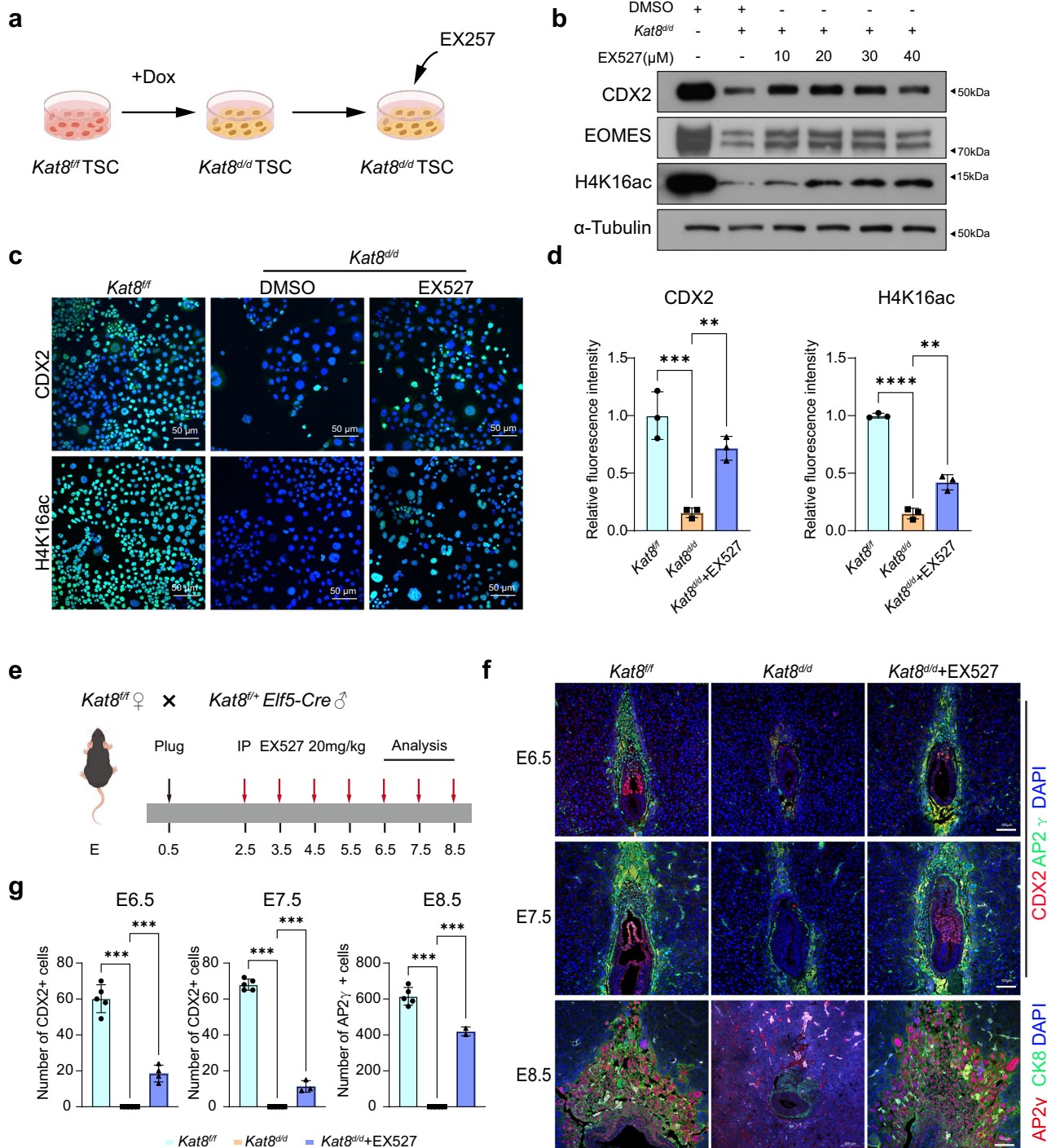

**Fig. 5 | EX527 partially rescue abnormal placental development caused by KAT8 knockout both in vitro and in vivo. a** Schematic representation illustrating the process of adding EX527 in *Kat8^{d/d}* mTSCs. The graphic elements were created by figdraw. **b** Immunoblot analysis of CDX2, EOMES and H4K16ac in *Kat8^{d/d}* mTSCs with or without EX527. **c** Immunofluorescent staining of CDX2 and H4K16ac in *Kat8^{d/d}* mTSCs with or without EX527. **d** The quantitative results of C. Data are representative of three independent experiments (*n* = 3) and values are normalized to *Kat8^{f/f}* control group and expressed in mean ± SEM. **P < 0.01, ***P < 0.001. Source data are provided as a Source Data file. **e** Schematic representation depicting the application of EX527 treatment on *Kat8^{d/d}* embryos. The graphic

elements were created by figdraw. **f** Immunofluorescent staining of CDX2, AP2γ and CK8 in *Kat8^{f/f}*, *Kat8^{d/d}* and EX527 treated *Kat8^{d/d}* embryos. **g** Quantification of CDX2+ cells or AP2γ+ cells in implantation sites of the indicated genotypes. Each dot in the column represents one implantation site (E6.5, *n* = 6 for *Kat8^{f/f}*, *n* = 6 for *Kat8^{d/d}*, *n* = 6 for *Kat8^{f/f}* + EX527; E7.5, *n* = 6 for *Kat8^{f/f}*, *n* = 6 for *Kat8^{d/d}*, *n* = 6 for *Kat8^{f/f}* + EX527; E8.5, *n* = 6 for *Kat8^{f/f}*, *n* = 6 for *Kat8^{d/d}*, *n* = 2 for *Kat8^{f/f}* + EX527). Two-tailed unpaired Student's *t*-test. Error bars, mean ± SEM. *P < 0.05, **P < 0.01, ***P < 0.001. (*n* = 3 mice for *Kat8^{f/f}*, *n* = 3 mice for *Kat8^{d/d}*, *n* = 3 mice for *Kat8^{f/f}* + EX527). Source data are provided as a Source Data file.

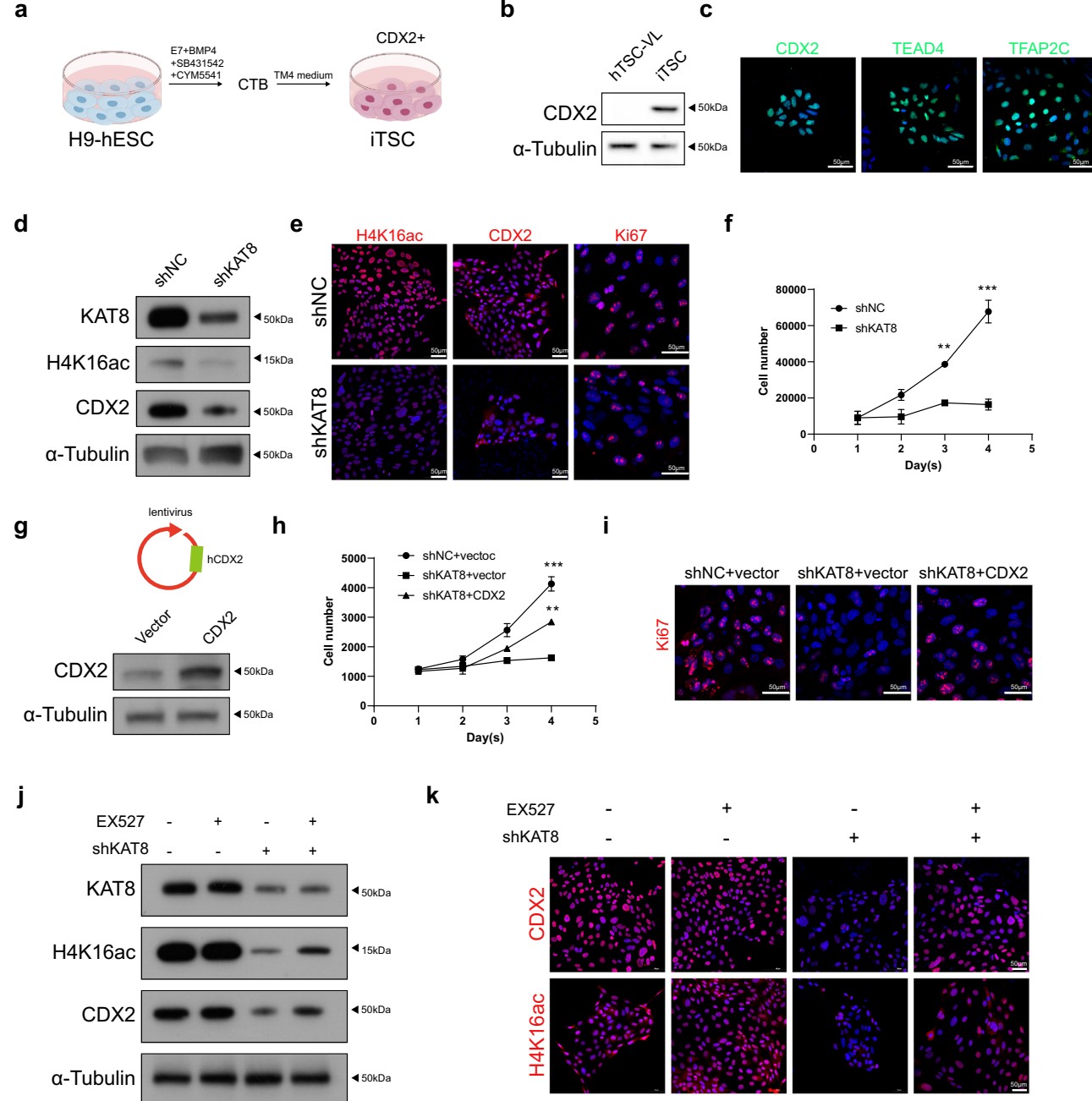

**Fig. 6 | KAT8-H4K16ac-CDX2 is conserved in human trophoblast cell.**
**a** Schematic representation illustrating the process of establishing iTSC. The graphic elements were created by figdraw. **b** Immunoblot analysis of CDX2 in hTSC from placenta villi and iTSC. **c** Immunofluorescent staining of CDX2, TEAD4, TFAP2C in iTSC. **d** Immunoblot analysis of KAT8, H4K16ac and CDX2 in iTSC and KAT8 knockdown iTSC. **e** Immunofluorescent staining of H4K16ac, CDX2, Ki67 in iTSC and KAT8 knockdown iTSC. **f** Cell counts of iTSC and KAT8 knockdown iTSC collected on indicated days. Data are representative of three independent experiments (*n* = 3). Two-tailed unpaired Student's *t*-test. Error bars, mean ± SEM. *P* = 0.005 (for D3), *P* = 0.0025 (for D4). Source data are provided as a Source Data

file. **g** Immunoblot analysis of CDX2 in iTSC and CDX2 overexpression iTSC. **h** Cell counts of iTSC and CDX2 overexpression iTSC collected on indicated days. Data are representative of three independent experiments (*n* = 3). Two-tailed unpaired Student's *t*-test. Error bars, mean ± SEM. *P* = 0.001 (for D4, between shNC and shKAT8), *P* = 0.0025 (for D4, between shNC and shKAT8 + CDX2). Source data are provided as a Source Data file. **i** Immunoblot analysis of Ki67 in iTSC, KAT8 knockdown iTSC and CDX2 overexpression iTSC. Data are representative of three independent experiments. **j** Immunoblot analysis of KAT8, CDX2, and H4K16ac in KAT8 knockdown iTSC with or without EX527. **k** Immunofluorescent staining of CDX2 and H4K16ac in KAT8 knockdown iTSC with or without EX527.

conservation of KAT8-mediated transcriptional regulation of CDX2 through H4K16ac in human trophoblast cells.

## The expression of KAT8 and H4K16ac is negatively correlated with RPL

To explore the potential function of KAT8-H4K16ac during placentation of human, we collected placental tissues at gestational

week 7, 19 and 37 and examined the expression pattern of KAT8 and H4K16ac. As shown in Fig. 7a, b, we revealed a gradual decrease in the expression of KAT8 and H4K16ac with the progression of pregnancy, indicating a lower expression of this axis in differentiated trophoblast cells. Notably, H4K16ac is predominantly localized in the CTB of control villous tissues, which is a subset of trophoblast stem cells capable of differentiating into other trophoblast cell subtypes,

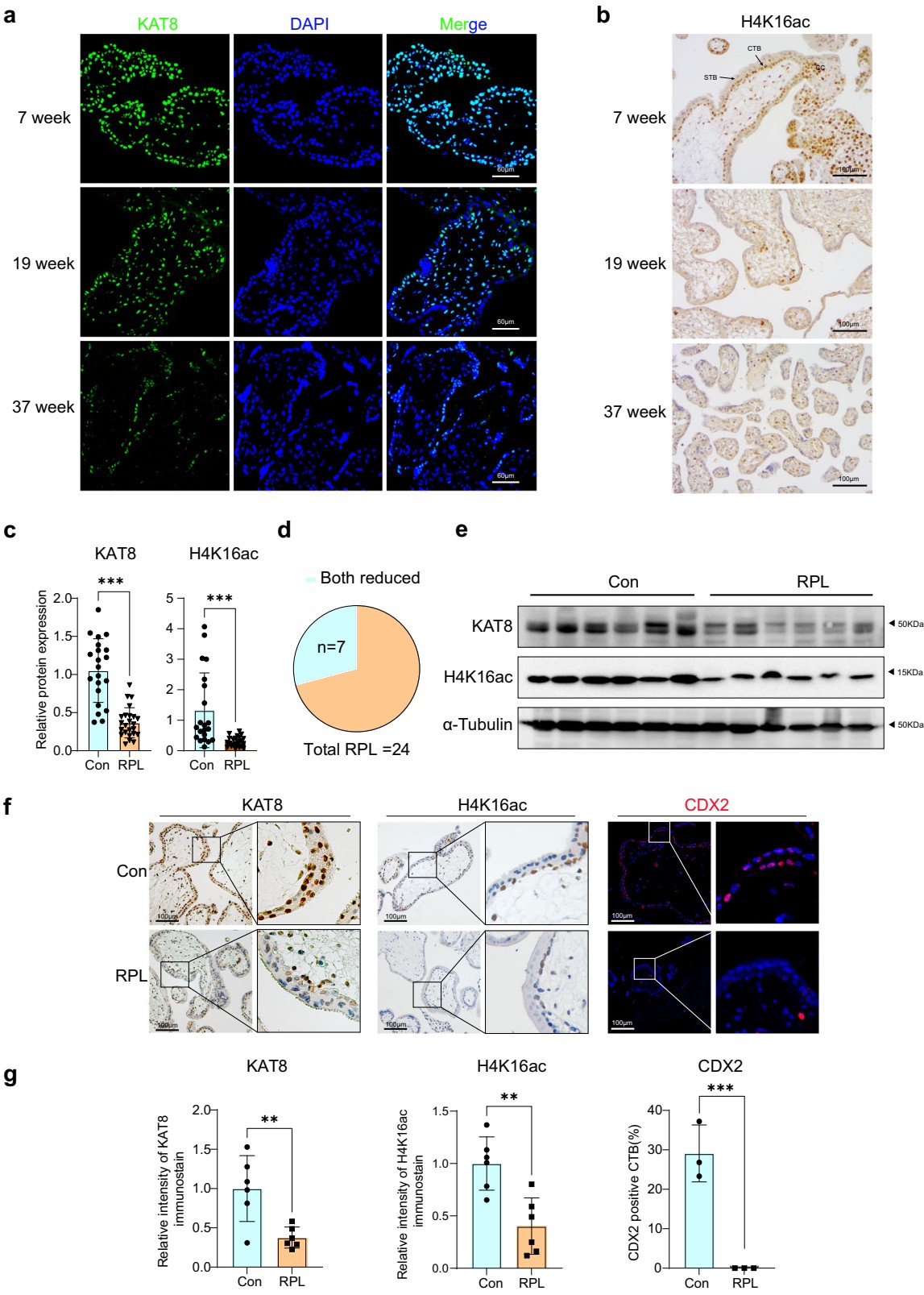

suggesting its potential role in promoting CTB proliferation and maintaining stemness. Given CDX2 undergoes rapid downregulation upon induction of differentiation in placenta trophoblast cells[13,38–40], we investigated the expression of CDX2 in the placenta villi of early stage of first trimester (6–7 week of pregnancy) placenta. Our results revealed that CDX2 was co-expressed in a subset of CTBs (TP63[+]) at 6 weeks of gestation (Supplementary Fig. 6a), yet the expression of

CDX2 was rarely remained in CTBs at 7 weeks (Supplementary Fig. 6a).

To assess the clinical relevance of KAT8-H4K16ac-CDX2 axis, we examined their protein expressions on villous tissues from a cohort of 24 patients diagnosed with unexplained RPL, excluding cases attributed to chromosomal abnormalities and uterine abnormalities, and from 28 age-matched healthy pregnancies of patients who voluntarily

**Fig. 7 | The expression of KAT8 and H4K16ac in placenta villous of normal pregnancies and RPL patients. a** Immunofluorescent staining of KAT8 in the placental villous tissues from normal pregnancies. **b** Immunohistochemical analysis of H4K16ac in the placental villous tissues from normal pregnancies. **c** The quantitative results of immunoblots of KAT8 and H4K16ac in the villi from normal ($n = 28$) and RPL ($n = 24$) pregnancies. α-Tubulin were used as loading controls and data are normalized to normal groups. Two-tailed unpaired Student's $t$-test. Error bars, mean ± SEM. $P < 0.0001$ (for KAT8), $P < 0.0001$ (for H4K16ac). Source data are provided as a Source Data file. **d** Statistical analysis of RPL cases ($n = 24$) exhibiting concurrent downregulation of KAT8 and H4K16ac ($n = 7$), as depicted in

immunoblot images of villous tissues obtained from normal and RPL placentas. **e** Representative immunoblot images of KAT8 and H4K16ac in the villi from normal ($n = 6$) and RPL ($n = 6$) placentas. **f** Representative staining for KAT8 (IHC), H4K16ac (IHC) and CDX2 (IF) in the villi from normal and RPL placentas. **g** Quantification of KAT8$^+$ cells, H4K16ac$^+$ and CDX2$^+$ cells in placenta villi from normal and RPL patients. Values are normalized to normal groups and expressed in mean ± SEM. Two-tailed unpaired Student's $t$-test, $*P < 0.05$, $**P < 0.01$, $***P < 0.001$. ($n = 6$ for KAT8, $n = 6$ for H4K16ac, $n = 3$ for CDX2). Source data are provided as a Source Data file.

opted for abortion (Supplementary Table S1). Our results revealed a significant decrease in the expression of KAT8 and H4K16ac in RPL placenta villus compared to the age matched control group (Fig. 7c and e). Among the 24 patients with RPL, a subset of 7 individuals exhibited concurrent downregulation of KAT8 and H4K16ac in the villous tissues (Fig. 7d). In line with the reduced protein level of KAT8-H4K16ac in RPL placenta, a consistent decrease of their expression levels in cytotrophoblasts of placenta villus in RPL patients were observed by IHC staining (Fig. 7f, g). Furthermore, compared to the control group, RPL patients exhibited a significant reduction in CDX2 expression in placental villi collected at 6 weeks of gestation (Fig. 7f, g). These findings indicate the significance of KAT8-H4K16ac-CDX2 axis in the maintenance of the early trophoblast linage in humans and the disruption of this axis is linked to RPL.

### EX527 ameliorates impaired self-renewal and growth in RPL patient-derived trophoblast organoids

In exploring the potential therapeutic avenue of targeting KAT8-H4K16ac-CDX2 axis for placenta-derived pregnancy loss, we generated human trophoblast organoids (hTOs) from first trimester placenta villi (human subjects were described in Methods)[6] (Supplementary Fig. 5a). To assess the fidelity of the hTOs, we performed H&E staining and IF staining, focusing on CTB markers AP2α and AP2γ, STB marker CGB, and epithelial marker CK7. As depicted in Supplementary Fig. 5b, these markers were positively stained in our hTOs system (Supplementary Fig. 5b), confirming that our organoid model faithfully recapitulates the features of its originating placenta tissue. We then examined the localization and expression of KAT8, H4K16ac, and CDX2 in hTOs. As shown in Supplementary Fig. 5c, we observed consistent expression patterns in hTOs when compared to their in vivo tissue counterparts. Specifically, KAT8 expressed in both CTB and STB, while H4K16ac showed predominant expression in CTB. Notably, CDX2 showed sparse expressed within CTB, consistent with previously observation[40].

To assess whether the supplementation of EX527 would mitigate KAT8 loss caused defects in hTOs, we first reduced KAT8 expression by using lentiviral infection of shKAT8 vector, and then supplemented EX527 alongside this knockdown. Subsequently, we seeded the cells into Matrigel to facilitate their growth into organoids (Fig. 8a). The knockdown efficiency was verified by RT-qPCR and western blot, which showed a marked decrease in KAT8 expression (Fig. 8b, c). Consistent with the findings observed in the iTSC results (Fig. 6k), the introduction of EX527 to the culture medium showed no discernible impact on the expression levels of CDX2 and H4K16ac (Supplementary Fig. 6b, c). Furthermore, there was no noticeable alteration in organoid growth, as evidenced by identical organoid diameters and organoid-forming efficiency (Supplementary Fig. 6d, e). Notably, the application of EX527 led to a significantly increase in the level of H4K16ac in KAT8 knockdown hTOs. Meanwhile, we observed the mRNA expression of CDX2 was also significantly increased (Fig. 8d). The elevation of H4K16ac and CDX2 upon EX527 treatment was reinforced by IF staining (Fig. 8e). The organoid forming efficiency and organoid diameter, indicators of stem cell self-renew capability and proliferation, were also detected. As shown in Fig. 8e, f, these results showed EX527 could effectively increase the organoid forming efficiency with enlarged

organoid size in hTOs of KAT8 knockdown, reinforcing that EX527 treatment effectively enhance the stemness and proliferation capability of hTOs.

To further affirm the therapeutic effect of EX527 in placentas exhibiting reduced expression of KAT8 and H4K16ac, we collected placental villi from three distinct RPL patients with low KAT8 and H4K16ac expression and generated hTOs (RPL-TOs) (Fig. 8g, h). As shown in Fig. 8i–k, RPL-TOs exhibited significantly reduced in the expression of CDX2, organoid formation ability compared to the control group, with limited passage ability up to the 4th generation (Fig. 8i–k). However, treatment with EX527 in RPL-TOs resulted in a significant increase in the diameter of formed organoids, the expression of CDX2 and the ability to pass over the 7th generation (Fig. 8i–k).

## Discussion

The irregular proliferation and differentiation of trophoblast cells during early gestation serve as the fundamental trigger for various pregnancy-related complications associated with the placenta, including recurrent pregnancy loss, which is defined as the occurrence of two or more consecutive spontaneous abortions within the first 20 weeks of gestation, with an approximate incidence rate of 2.6%[41–47]. Unraveling the intricacies of these underlying mechanisms may pave the way for innovative interventions to address this disorder. Here, we present compelling genetic evidence demonstrating the essential role of KAT8-mediated H4K16ac in trophoblast stemness and CDX2 expression. Increasing H4K16ac levels with the SIRT1 inhibitor, EX527, restores CDX2 levels and enhances placenta development in trophoblast-specific *Kat8* knockout mouse models. Clinically, we observed that reduced KAT8 and H4K16ac expression are associated with RPL. These patients exhibited impaired TSC self-renewal and reduced capabilities for organoid forming and passaging. Remarkably, the application of EX527 significantly enhances stemness in trophoblast organoids derived from these RPL patients. Our findings suggest that targeting the KAT8-H4K16ac-CDX2 axis could be a promising therapeutic strategy for placenta-derived pregnancy loss.

Although our CUT&Tag results showed that a substantial portion (>70%) of genetic loci peaks exhibited concurrent presence of both KAT8 and H4K16ac, KAT8 also affects certain genes independently of H4K16ac. Functionally, KAT8 serves as a catalytic subunit in two distinct protein complexes in both Drosophila and mammals: the male-specific lethal (MSL) and the non-specific lethal (NSL) complexes[48–51]. These MSL and NSL complexes possess shared regulatory functions as well as distinct roles in governing pluripotency and differentiation processes. KAT8 catalyzes the acetylation of H4K16 as part of the MSL complex, which predominantly binds to gene bodies and is enriched at stem cell-specific genes[26]. On the other hand, KAT8 could also catalyzes the acetylation of H4K5 and H4K8 as part of the NSL complex, exclusively binding to promoters, and being crucial for cell proliferation and survival[52]. This dualistic role of KAT8 likely accounts for the inadequacy of relying solely on H4K16ac activation or CDX2 overexpression to fully rescue the phenotype associated with *Kat8* knockout. Intriguingly, our investigations consistently unveiled a pronounced enrichment of terms associated with DNA damage and cellular senescence in *Kat8* knockout TSCs. Therefore, it would be of

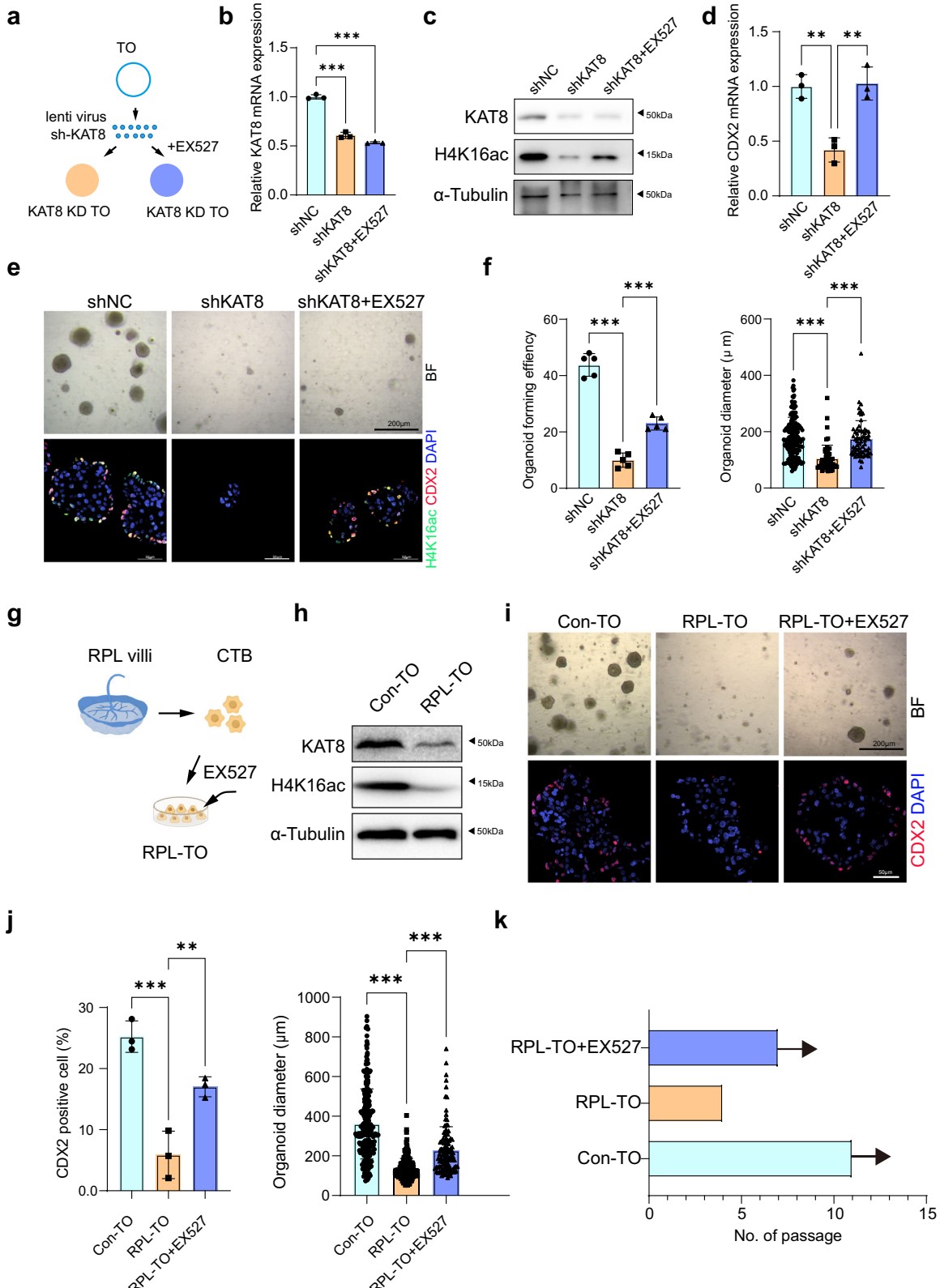

substantial interest to delve deeper into the regulatory mechanisms underlying these genes and determine whether they are governed by the NSL complex.

SIRT1 and KAT8 have coordinate actions in regulating H4K16ac-mediated gene expression[53]. SIRT1 was consistently detected in syncytiotrophoblast, a differentiated cell type in placenta, throughout the entirety of normal gestation. The deletion of *Sirt1* resulted in smaller

placentas with diminished junctional zones and less mature labyrinth structures[33]. Specifically deleting *Sirt1* in trophoblast cells using *Elf5-Cre* further emphasizing its pivotal role in facilitating the proper terminal differentiation of trophoblast cells, while its absence in TSCs notably hindered differentiation[54]. Notably, we observed the expression of H4K16ac is lower in syncytiotrophoblast than cytotrophoblast, and the expression is reduced alongside placental differentiation,

**Fig. 8 | EX527 treatment effectively enhance the stemness and proliferation capability of hTOs derived from RPL patients. a** Schematic representation illustrating the application of EX527 treatment on human trophoblast organoids (hTOs) following knockdown of KAT8 (KAT8-KD) using lentiviral vectors. **b** Quantitative real-time PCR analysis of KAT8 mRNA in KAT8-KD hTOs with or without EX527. The values are normalized to ACTB. Data are representative of five independent experiments ($n = 3$). Two-tailed unpaired Student's $t$-test. Error bars, mean ± SEM. $P < 0.0001$ (between shNC and shKAT8), $P < 0.0001$ (between shNC and shKAT8 + EX527). Source data are provided as a Source Data file. **c** Immunoblot analysis of KAT8 and H4K16ac in KAT8-KD hTOs with or without EX527. **d** Quantitative real-time PCR analysis of CDX2 mRNA in KAT8-KD hTOs with or without EX527. The values are normalized to ACTB. Data are representative of three independent experiments ($n = 3$). Two-tailed unpaired Student's $t$-test. Error bars, mean ± SEM. $P = 0.0029$ (between shNC and shKAT8), $P = 0.005$ (between shKAT8 and shKAT8 + EX527). Source data are provided as a Source Data file. **e** Brightfield images and IF images of CDX2 and H4K16ac in KAT8-KD hTOs with or without EX527. **f** The quantification results of organoid forming efficiency and diameter for hTOs in indicated groups. For organoid forming efficiency, data are representative of five independent experiments ($n = 5$). Two-tailed unpaired Student's $t$-test. Error bars, mean ± SEM. $P < 0.001$ (between shNC and shKAT8), $P < 0.001$ (between

shKAT8 and shKAT8 + EX527). For diameter, each dot in the column represents one TO ($n = 155$, shNC; $n = 56$, shKAT8; $n = 65$, shKAT8 + EX527). Two-tailed unpaired Student's $t$-test. Error bars, mean ± SEM. $P < 0.001$ (between shNC and shKAT8), $P < 0.001$ (between shKAT8 and shKAT8 + EX527). Source data are provided as a Source Data file. **g** Schematic illustrating the collection of RPL placental tissue, generation of RPL-derived trophoblast organoids (RPL-TOs), and continuous treatment with EX527, $n = 3$. The graphic elements were created by figdraw. **h** Immunoblot analysis of KAT8 and H4K16ac in the villi from normal and RPL pregnancies. **i** Brightfield images and immunofluorescent staining of CDX2 in RPL-TOs with or without EX527. **j** The quantification results of CDX2+ cells and diameter for RPL-TOs with or without EX527. For CDX2$^+$ cells, data are representative of three independent experiments ($n = 3$). Two-tailed unpaired Student's $t$-test. Error bars, mean ± SEM. $P = 0.002$ (between Con-TOs and RPL-TOs), $P = 0.0101$ (between RPL-TOs and RPL-TOs + EX527). For diameter, each dot in the column represents one TO ($n = 249$, Con-TOs; $n = 208$, RPL-TOs; $n = 117$, RPL-TOs + EX527). Two-tailed unpaired Student's $t$-test. Error bars, mean ± SEM. $P < 0.001$ (between Con-TOs and RPL-TOs), $P < 0.001$ (between RPL-TOs and RPL-TOs + EX527). Source data are provided as a Source Data file. **k** Analysis of the passage number for RPL-TOs with or without EX527 treatment.

suggesting that H4K16ac plays a critical role in maintaining TSC stemness and promoting TSC proliferation. In line with this perspective, the trophoblast-specific knockout of *Kat8* causes a relative earlier phenotype with embryo lethality right postimplantation. The phenotype is characterized by the absence of the trophoblast lineage, largely due to the loss of H4K16ac and the TSC stemness marker CDX2, thus highlighting the importance of H4K16ac in TSC maintenance and proliferation during early gestational stage. However, we're not excluding the KAT8 might also critical for TSC differentiation, as the key related genes such as *Syna*, *Ctsq* and *Gcm1* etc. are also affected.

More than 50% of RPL cases fall under the category of unexplained recurrent pregnancy loss, indicating that no specific causative factor can be identified through diagnostic evaluations. While various treatments are available for these patients, there is no universal consensus on the most effective approach. Our study sheds light on a potential therapeutic avenue for a subgroup of RPL patients with diminished H4K16ac in trophoblast cells, through the administration of EX527. Notably, EX527 has already entered multiple clinical trials (NCT04184323, NCT01521585) for endometriosis, progesterone resistance and Huntington's disease. These clinical investigations have demonstrated the safety and tolerability of EX527 in human subjects, with no significant adverse reactions reported. This favorable safety profile enhances the translational potential of EX527 for the treatment of placenta-derived disorders. Our preclinical experiments conducted on RPL organoids further support the effectiveness of EX527 in enhancing trophoblast self-renewal and growth, albeit within a limited patient cohort. Consequently, EX527 offers a promising avenue for the treatment of a specific subset of RPL patients.

In conclusion, our study provides insights into the regulatory mechanisms underlying trophoblast development and its implications in RPL. We identified KAT8 as a crucial regulator of trophoblast stem cell proliferation and self-renewal, and demonstrated its role in activating CDX2 expression through H4K16ac. Targeting the KAT8-H4K16ac in trophoblast shows potential as a therapeutic strategy to mitigate placenta-derived pregnancy loss.

## Methods
### Mice
*Elf5-Cre* transgenic mice were generated using the previously reported method[29]. *Kat8^f/f^* mice were generated by inserting Loxp sequences on both sides of exons 2 and 3 of the *Kat8* gene. The method for specific knockout of *Kat8* in trophoblast cells involved crossing 8 week-old *Kat8^f/f^* female mice with *Kat8^f/+^;Elf5-Cre* male mice. According to Mendelian inheritance laws, approximately 1/4 of the embryos achieved

specific knockout of *Kat8* in trophoblast cells. Mice with vaginal plugs were considered embryonic day 0.5 (E0.5). All mice utilized in this study belonged to the C57BL6 strain. They were housed in a specific pathogen-free environment, maintaining a 12 h light/dark cycle, with controlled temperature (20–25 °C) and humidity (50–70%). Food and water were available ad libitum, and housing was provided in the Animal Care Facility at the National Institute of Biological Sciences, Beijing. This study was approved by the Animal Care Committee of National Institute of Biological Sciences.

### Mouse trophoblast stem cell derivation, culture
The derivation and maintenance of mTSCs were performed as previously described[55,56]. Briefly, *Kat8^f/f^* female mice were co-housed with *Kat8^f/f^* male mice, and blastocysts were collected at E3.5 and placed in a culture dish pre-coated with mitomycin C-treated mouse embryonic fibroblast (MEF) feeders. The TS + F4H medium, consisting of RPMI 1640 supplemented with 20% FBS, 2 mM L-glutamine (Life Technologies, 25030-081), 1 mM sodium pyruvate (Gibco, 11360070), 100 μM β-mercaptoethanol (Gibco, 21985023), 25 ng/ml FGF4 (Peprotech, 100-31), and 1 mg/ml heparin (Sigma, H3149), was added to the dish. When the outgrowth of blastocysts reached the desired size, it was dissociated into single cells using 0.25% trypsin/EDTA and then cultured in MEF-CM medium, which contained 70% MEF-conditioned medium, 30% TS medium, and 1.5X FGF4/heparin. Clones of TSCs were formed during the subsequent culture. The stable proliferating TSCs were then transferred to a serum-free culture system[56], where the culture dish was pre-coated with Matrigel. The TXM medium, consisting of Advanced DMEM/F12 supplemented with 1% penicillin-streptomycin, 2 mM L-glutamine, non-essential amino acids (Gibco, 11140050), 64 mg/L L-ascorbic acid-2-phosphate magnesium (Sigma, A8960), 0.05% BSA, 1 mM sodium pyruvate, 0.5% B27 (Life Technologies, 17502048), 1% ITS supplement (Wako, 094-06761), 100 μM β-mercaptoethanol, 25 ng/ml FGF4, 1 mg/ml heparin, 2 ng/ml TGF-β (Peprotech, 100-21), 5 μM Y27632 (Wako, 030-24021), and 200 nM ZSTK474 (Selleck, S1072), was used for the culture.

### Human samples
Samples of villous tissue from patients with RPL and those healthy women who terminated pregnancy for other reasons were collected from the Third Affiliated Hospital of Guangzhou Medical University from September 1, 2021, to October 2, 2023. Patients presenting with the following conditions were systematically excluded from the study cohort: (1) congenital anomalies affecting the reproductive system; (2) aberrant karyotypes observed in both parents and abortuses; (3)

endocrine or metabolic dysfunctions; (4) autoimmune disorders; (5) coexisting major illnesses; (6) inadequate pharmaceutical intervention, as well as exposure to hazardous chemicals or radiation. This study has been approved by the Ethics Committee of the Third Affiliated Hospital of Guangzhou Medical University, with approval number 2018002, and informed consent was obtained from all participants. This study was conducted in accordance with the Declaration of Helsinki. Recurrent pregnancy loss was defined as two or more unexplained miscarriages according to the criteria set by the Practice Committee of the American Society for Reproductive Medicine[41]. The clinical characteristics of the patients included in this study have been summarized in Table S1. After collection, the chorionic tissue was rinsed with ice-cold PBS to remove residual blood and divided into several portions. Some portions were fixed in 4% paraformaldehyde for immunohistochemical analysis, while others were rapidly frozen in liquid nitrogen for protein and RNA extraction.

## Derivation and culture of iTSC cell line

H9-hESCs is a gift from Prof. Yong Fan and Dr. Chaohui Li in Guangzhou Medical University. We validated the H9-hESCs reliability using STR profiling. H9-hESCs were cultured in mTeSR Plus medium (Stem cell technologies, 100-0276). On the 2nd day of passage, H9-hESCs were treated with 2 µM CYM5541 (Selleck, S6552), 25 µM SB431542 (Wako, 031-24291), and 20 ng/ml BMP4 (Thermo Fisher, PHC9534) in TeSR-E7 medium (Stem cell technologies, 05914) for 3 days, with daily medium changes. After 3 days, the cells were passaged onto culture dishes precoated with 3 µg/ml of vitronectin (Thermo Fisher, A14700) and 1 µg/ml of Laminin 521 (Stem cell technologies, 77003). Subsequently, the cells were switched to TM4 medium, TeSR-E6 medium (Stem cell technologies, 05946) supplemented with 2 µM CYM5541, 0.5 µM A83-01 (Wako, 035-24113), 25 ng/ml FGF10 (Stem cell, 78037), and 2 µM CHIR99021 (Wako, 038-23101), for maintaining the CDX2$^+$ trophoblast stemness status.

## Derivation and culture of human trophoblast organoids

Placental villi in early pregnancy (≤6 weeks) were dissected into small fragments and subjected to enzymatic digestion using a solution comprising TrypLE and Accumax (at a 1:1 ratio) for 20 min at 37 °C. The resulting cell suspensions were subsequently filtered through a 70 mm mesh filter. To isolate CTBs, immunomagnetic purification was performed using the EasySep PE selection kit (Stemcell, 17684) along with a PE-conjugated anti-CD49f antibody (Miltenyi, 130-119-807). The purified CTBs were then seeded onto Matrigel droplets (50 µl) placed at the center of wells in a 24-well plate. The plates were inverted and incubated for 15 min at 37 °C, after which they were overlaid with trophoblast organoid medium (TOM)[57]. The medium composition included Advanced DMEM/F-12 (Gibco, 12634010), 100 µg/ml Primocin (Invivogen, ant-pm-1), 1.25 mM N-acetyl-L-cysteine (Sigma, A9165), 1X B27 supplement (Life Technologies, 12587010), 1X N2 supplement (Life Technologies, 17502048), 2 mM L-glutamine (Life Technologies, 25030-081), 2 µM Y-27632 (Wako, 030-24021), 2.5 µM PGE2 (Sigma, P0409), 2.5 µM A83-01 (Wako, 039-24111), 1.5 µM CHIR99021 (Wako, 038-23101), 50 ng/mL human recombinant HGF (Peprotech, 100-39), 80 ng/mL human recombinant R-spondin (Proteintech, HZ-1328), 100 ng/mL human recombinant FGF2 (Peprotech, 100-18), 50 ng/mL human recombinant EGF (Wako, 053-07871). The medium was refreshed every 2-3 days. For passaging, the organoids were released from the Matrigel by incubating them with ice-cold cell recovery solution for 20 min on ice. Subsequently, the organoids were transferred to 15 ml tubes and centrifuged at 1000 g. The supernatant was carefully removed, and the cell pellets were further digested using Stem Pro Accutase (Stemcell, A1110501) at a temperature of 37 °C for a duration of 5 min.

For lentiviral transduction, organoids were dissociated and subjected to "spinoculation". In brief, viral supernatants were added to cells in ultra-low 24-well plates, followed by centrifugation at 600 g for 60 min at 37 °C. The cells were then incubated at 37 °C for an additional 2–4 h, after which the supernatants were collected and the cells were re-seeded in media containing Matrigel. Infected organoids were selected 72 h post-viral transduction using puromycin, as indicated.

## Western blot analysis

Protein extraction, protein concentration determination, and Western blot analysis were performed following previously established protocols[46]. Briefly, tissues or cells were lysed on ice for 20 min using protein lysis buffer (Beyotime, P0013) supplemented with protease inhibitors (Roche). The lysates were then centrifuged at 14,000 g for 20 min at 4 °C, and the supernatants were collected. Protein concentration was determined using the BCA Protein Assay Kit (Beyotime, P0009) according to the manufacturer's instructions. SDS-PAGE gels were prepared, and proteins were separated at a constant voltage of 100 V. Subsequently, proteins were transferred onto PVDF membranes (Millipore, USA) using a semi-dry transfer system, and the membranes were incubated with primary antibodies for KAT8 (Abcam, ab200660) (1:1000), H4K16ac (Millipore, 07-329) (1:1000), CDX2 (Abcam, ab76541) (1:1000), EOMES (Abcam, ab23345) (1:1000), α-Tubulin (Proteintech, 11224-1-AP) (1:10000) overnight at 4 °C. After washing, the membranes were incubated with HRP-conjugated goat anti-rabbit IgG (Sigma, A6154) (1:5000) or HRP-conjugated goat anti-mouse IgG (Sigma, A4416) (1:5000) at room temperature for 1 h. Protein bands were visualized using an ECL chemiluminescent substrate (AQ, AQ529) and captured on X-ray film. All uncropped and unprocessed scans of western blots were supplied in Supplementary Fig. 7 in the Supplementary Information.

## Immunofluorescent staining analysis

The isolated implantation sites and villous tissue were fixed in 4% paraformaldehyde, with the fixation time dependent on the size of the tissue. After fixation, the tissues were dehydrated, with the dehydration time also dependent on tissue size, followed by embedding in paraffin. Tissue sections with a thickness of 5 µm were used for immunohistochemistry, immunofluorescence, and H&E staining. For immunofluorescence, after dewaxing and rehydration of the sections, high-pressure antigen retrieval was performed using citrate buffer, followed by blocking with 0.5% BSA at room temperature for 1 h. The primary antibody KAT8 (Abcam, ab200660) (1:100), H4K16ac (Millipore, 07-329) (1:500), Ki67 (Invitrogen, 14-5698-82) (1:500), γH2A.X (Millipore, 05-636) (1:100), CDX2 (Abcam, ab76541) (1:100), EOMES (Abcam, ab23345) (1:100), AP2γ (Santa Cruz, SC12762) (1:100), CK8 (DSHB, AB_531826) (1:100), AP2α (Abcam, ab108311) (1:100), CGB (Abcam, ab9582) (1:100) was incubated overnight at 4 °C. Subsequently, Alexa Fluor® 488 conjugated donkey anti-mouse IgG (Invitrogen, A21202) (1:500), Alexa Fluor® 488 conjugated donkey anti-rabbit IgG (Invitrogen, A21206) (1:500), Alexa Fluor® 546 conjugated donkey anti-mouse IgG (Invitrogen, A10036) (1:500), or Alexa Fluor® 546 conjugated donkey anti-rabbit IgG (Invitrogen, A10040) (1:500) were applied at room temperature for 1 h, followed by DAPI staining and mounting with anti-fade mounting medium. For immunohistochemistry, after high-pressure antigen retrieval, endogenous peroxidase activity was quenched using a 3% hydrogen peroxide/methanol solution, and DAB staining was performed using an HRP-conjugated secondary antibody.

## Plasmid construction and plasmid transfection

To generate the full-length coding region of mouse *Cdx2* and *Kat8* from mTSCs complementary DNA. The amplified *Cdx2* sequence, along with the *Kat8*, E350Q-*Kat8* and *Cre* sequence, was then cloned into the PiggyBac vector (Primers in Table S2). For plasmid transfection, mTSCs were seeded in culture dishes 1 day prior to transfection and allowed to adhere overnight. When the cells reached 80%

confluency, the PiggyBac vector containing the *Cre*, *Kat8* and *Cdx2* sequences was transfected into the cells using the 4D-Nucleofector X unit transfection system (Lonza, USA) with the CG-104 electroporation program. After electroporation, 500 μl of TXM medium was added to the electroporation cuvette and incubated at room temperature for 5 min. The mixture was then transferred to a culture dish, and fresh TXM medium was replaced after 8 h. After 48 h, the corresponding selection drugs were added for pressure selection, and stable transgenic colonies were picked for further experiments.

To generate lentiviral vectors, including hCDX2 (Primers in Table S2), shKAT8 and shNC (from Sigma shRNA library, TRCN0000034875), HEK293T cells were co-transfected with lentiviral vectors, psPAX2, and pMD2.G at a ratio of 10:7.5:2.5 using jetPRIME Transfection Reagent, following the manufacturer's instructions. Media were replaced 8 h post-transfection, and viral supernatants were collected 48 h later after passing through a 0.45 μm filter. The collected supernatants were aliquoted and stored at −80 °C.

### qRT-PCR
Total RNA was extracted from mTSCs using TRIZOL reagent. For reverse transcription, 1 μg of RNA was used with the PrimeScript™ RT Reagent Kit (TAKARA, RR047A) following the manufacturer's protocols. The expression levels of different genes were detected using TB Green® Premix Ex Taq™ II (TAKARA, RR820A) on the ABI QuantStudio 5 Real-Time PCR system. The expression levels of all genes were normalized to ACTB. The specific primer sequences used for PCR amplification can be found in Table S3.

### RNA-seq and data analysis
Total RNA from mTSCs was extracted using TRIZOL reagent. Following library preparation and pooling of different samples, the samples were subjected to Illumina sequencing. The raw data (raw reads) obtained from Illumina sequencing were initially processed using in-house Perl scripts. All downstream analyses were performed using the clean data without rRNA. Differential expression analysis was conducted using EdgeR. The resulting *p*-values were adjusted using the Benjamini and Hochberg's approach to control the false discovery rate. Genes ($P < 0.05$, $Log_2FC \geq 1$) were considered to be differentially expressed.

### CUT&Tag and data analysis
CUT&Tag Experiment using NovoNGS® CUT&Tag® 4.0 High-Sensitivity Kit (Novoprotein, N259-YH01) was performed according to the manufacturer's instructions. About 100,000 cells were centrifuged at 1000 g for 5 min at room temperature and resuspended in Dilution buffer. An appropriate amount of ConA magnetic beads (10 μL per sample) was added to the binding buffer and washed once in the same buffer. Each time, the beads were placed on a magnetic rack to separate them from the buffer and then resuspended in binding buffer. The liquid was removed using a magnetic rack, and the cells were added to the beads. The mixture was incubated at a slow speed and room temperature for 10 min to allow the cells to bind to the beads. After cell-bead binding, the beads were collected using a magnetic rack, and the diluted primary antibody against KAT8 and H4K16ac was added and incubated overnight at 4 °C. The supernatant was removed, and the diluted secondary antibody was added and incubated at slow speed and room temperature for 1 h. The supernatant was removed, and the diluted transposase-containing buffer was added and incubated at slow speed and room temperature for 1 h to allow transposase-cell binding. The supernatant was removed, and the buffer containing $MgCl_2$ was added. The mixture was incubated at slow speed and 37 °C for 1 h to fragmentize the DNA. To stop the fragmentation reaction, 5 μL of Stop Buffer and 1 μL of Proteinase K (optional) were added to the incubated samples. The samples were vortexed at high speed and incubated at 55 °C in a metal bath for 10 min. Then, 2 times the volume of DNA Extract Beads (-144 μL) was added to the terminated samples, mixed by pipetting, and incubated at room temperature for 5 min to extract the DNA fragments. The extracted DNA fragments were used as templates for PCR amplification and library construction. The i5 primer, i7 primer, template, and PCR enzyme from the kit were added, and the number of PCR cycles was determined based on the cell quantity and target protein selection. The PCR products were purified using NovoNGS® DNA Clean Beads. After purification, the library fragment distribution was analyzed using the Agilent Technologies 2100 Bioanalyzer. The sequencing was performed using the Illumina Novaseq-6000 system. Following the principle of equimolar pooling based on the ratio of material quantity to data quantity, the mixed libraries were diluted to the desired sequencing concentration. After denaturation into single-stranded DNA using sodium hydroxide, the libraries were mixed with the reagents from The Novaseq Xp 4-Lane Kit 1.5. The mixture was then loaded into the sequencing instrument for sequencing.

### Figure preparation
The main figures were assembled in Adobe Illustrator; all other figures were assembled in Adobe Photoshop. The cartoon elements in Figs. 1c, 2a, 4g, 5a, 5e, 6a, 8g and supplementary Fig. 5a were custom-created with authorization from figdraw (www.figdraw.com). The elements utilized in Figs. 1b, 8a and 3d were hand-drawn using Adobe Illustrator.

### Statistics and reproducibility
For all histology, immunofluorescence, western blot, and qPCR experiments, we performed at least three independent biological replicates. All uncropped and unprocessed scans of the western blots were supplied in Supplementary Fig. 7 in the Supplementary Information.

### Statistical analysis
Statistical analysis was conducted using the GraphPad Prism software. The comparison of means was assessed using the independent-samples Student *t*-test. The data are presented as means ± SEM.

### Reporting summary
Further information on research design is available in the Nature Portfolio Reporting Summary linked to this article.

## Data availability
All sequencing data generated in this study have been deposited in the Genome Sequence Archive (GSA) (https://ngdc.cncb.ac.cn/gsa/) of China National Center for Bioinformation-National Genomics Data Center (CNCB-NGDC) under accession code: CRA015385, CRA015387, CRA015390. Source data are provided with this paper.

## Code availability
The custom scripts used in this study will be available on request from the corresponding authors.

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

## Acknowledgements

We thank Prof. Yong Fan and Dr. Chaohui Li from Guangzhou Medical University for providing H9-hESCs. S.B., L.H. and Y.C. contributed equally to this work. Work on this project was supported by grants from the National Key Research and Development Program of China (2022YFC2704500 to D.C., 2022YFC2702501 to S.Z. and H.W.), the Key Program of National Natural Science Foundation of China (81830045 to D.C.), the National Natural Science Foundation of China (82071652 to L.D., 82371674 to L.D., 82171666 to D.C., 82201861 to S.B., 82271695 to Z.T., 31701014 to Y.C., 82288102 to H.W.), China Postdoctoral Science Foundation (2021M700945 to S.B.), the Mobility program of Sino German Center (M-0586 to S.Z. and J.K.), the Science and Technology Program of Guangzhou (202201020573 to S.Z., 2023A03J0378 to Z.T., 2022A1515110166 to S.B.), General Project of Guangzhou Science and Technology Bureau (202201010907 to W.S.), General Guidance Project of Guangzhou Municipal Health Commission (20231A011094 to W.S.), Beijing Maternal and Child Health Care Hospital 'Discipline Backbone' Plan Special Funds (XKGG201904 to Y.C.), Beijing Natural Science Foundation (7242054 to Y.C.).

## Author contributions

Conceptualization: S.Z., F.W., S.B., Y.C. Methodology: S.B., L.H., S.L., B.H., L.Z., Y.H., B.D., D.X., Y.W. (Yijing Wang), W.S., Z.H., X.X. Investigation: S.B., L.H., Y.C. Visualization: S.Z., S.B., L.H., Y.W. (Yifan Wang), Z.H. Funding acquisition: S.Z., S.B., Z.T., D.C., Y.C., W.S. Supervision: H.W., D.C., F.W., S.Z. Writing original draft: S.B., S.Z. Writing, review & editing: S.Z., Z.H., L.D., Z.T., J.K.

## Competing interests

The authors declare no competing interests.

## Additional information

[1]Department of Obstetrics and Gynecology, The Third Affiliated Hospital of Guangzhou Medical University, Guangzhou 510150, China. [2]Guangdong Provincial Key Laboratory of Major Obstetric Diseases, The Third Affiliated Hospital of Guangzhou Medical University, Guangzhou 510150, China. [3]Guangdong Provincial Clinical Research Center for Obstetrics and Gynecology, The Third Affiliated Hospital of Guangzhou Medical University, Guangzhou 510150, China. [4]Guangdong-Hong Kong-Macao Great Bay Area Higher Education Joint Laboratory of Maternal-Fetal Medicine, The Third Affiliated Hospital of Guangzhou Medical University, Guangzhou 510150, China. [5]Central Laboratory, Beijing Obstetrics and Gynecology Hospital, Capital Medical University. Beijing Maternal and Child Health Care Hospital, Beijing 100026, China. [6]National Institute of Biological Sciences, Beijing 102206, China. [7]Tsinghua Institute of Multidisciplinary Biomedical Research, Tsinghua University, Beijing 102206, China. [8]Institute of Transfusion Medicine and Immunology, Mannheim Institute of Innate Immunosciences (MI3), Medical Faculty Mannheim, Heidelberg University, 68167 Mannheim, Germany. [9]German Red Cross Blood Service Baden-Württemberg-Hessen, 68167 Mannheim, Germany. [10]Laboratory of Translational Cellular and Molecular Biomedicine, National Research Tomsk State University, Tomsk, Russia. [11]Fujian Provincial Key Laboratory of Reproductive Health Research, Department of Obstetrics and Gynecology, The First Affiliated Hospital of Xiamen University, School of Medicine, Xiamen University, Xiamen 361102, China. [12]These authors contributed equally: Shilei Bi, Lijun Huang, and Yongjie Chen. ✉e-mail: haibin.wang@vip.163.com; gzdrchen@gzhmu.edu.cn; wangfengchao@nibs.ac.cn; shuang1zhang@gzhmu.edu.cn

