## [Peer Review File · Nature Communications]

KAT8-mediated H4K16ac is Essential for Sustaining Trophoblast Self-renewal and Proliferation via Regulating CDX2Reviewers' Comments:

Reviewer #1:

Remarks to the Author:

General Comment:

The authors have elucidated the role of KAT-8 and H4K16ac in maintaining stem cell properties during placental development, using genetically modified mice, mouse TS cells, human placental tissues, and placental organoids. The mouse data is particularly conclusive, demonstrating the critical role of KAT8 in embryonic development through sophisticated experimental methods like gene modification and rescue experiments. The significance of these findings is noteworthy. The potential usefulness of EX527 treatment is also suggested, which could be of great importance. However, the human data seem less robust compared to the mouse data. If human data are to be included, improvement in this section is necessary.

Major Comments:

1. Recent RNA-seq studies suggest that CDX2 is not expressed in the first-trimester placenta (Vento-Tormo et al., Nature 2018; Suryawanshi et al., Science Adv 2018). The paper discusses relative expression levels using qPCR, but isn't the absolute expression of CDX2 quite low?
2. While some papers report strong CDX2 staining in cytotrophoblast cells, most use the same monoclonal antibody (EPR2764Y) as in this study. It would be advisable to use other well-validated antibodies for detecting CDX2 in the human placenta.
3. The proportion of CDX2 positive cells in Fig. 7J is quite low at 0.25%, raising questions about the accuracy and credibility of the expression.
4. The paper mentions a KAT8-H4K16ac-CDX2 axis in humans, but there is no proof that CDX2 controls the proliferation of organoids. Is the expression of CDX2 too low or non-existent? For the human placental organoid section, comprehensive expression analysis similar to what was done for mice would be beneficial, particularly regarding KAT8 knockdown and RPL-TSO.

Minor Comments:

1. Line 86: A reference is needed for "Extensive research has elucidated the pivotal role of H4K16ac in maintaining the pluripotency of embryonic stem cells (ESCs)."
2. Ref. 16: Indent confirmation is needed.
3. Line 108: More detail on the staining results in Fig. 1A would be beneficial, especially for understanding the localization of KAT8 at E9.5. Adding annotations in the figure to indicate the magnified areas could improve clarity.
4. Line 117: The expression "loss of Kat8d/d" seems redundant.
5. Line 137: For Kat8d/d mice, "significantly reduced" is stated, but there is no statistical significant difference between the two types of mice.
6. Line 238: The increase in CDX2 does not appear to be dose-dependent.
7. Line 294: A citation is needed.
8. In Fig. 7K and Sup Fig. 5A, "Trophoblast stem cells" should be labeled as "Cytotrophoblast," as the organoids do not seem to be "Stem cell-derived" based on the methods in this study.

9. Line 375: The statement "This decline in H4K16ac correlates negatively with SIRT1 expression" needs clarification. Is this observed in the current study? If based on previous reports, a citation is required.

Reviewer #2:

Remarks to the Author:

The study presented by the authors investigates the role of Kat8, a MYST family histone acetyltransferase, in regulating trophoblast self-renewal and differentiation during early gestation. The findings demonstrate that trophoblast-specific deletion of Kat8 results in extraembryonic ectoderm abnormalities and embryonic lethality. Through comprehensive analyses, including RNA-seq and CUT&Tag, the study uncovers KAT8's role in the transcriptional activation of the trophoblast stemness marker CDX2, mediated by acetylating H4K16. The rescue experiments with CDX2 overexpression and the use of the SIRT1 inhibitor, EX527, provide valuable insights into the regulatory mechanisms. Clinical correlation with recurrent pregnancy loss (RPL) and the amelioration of trophoblast organoid abnormalities with EX527 treatment further strengthen the significance of the study. However, additional experiments and thorough validation are recommended to strengthen the causal relationships proposed in the manuscript.

1. The author should conduct research on the expression changes of CDX2, including mRNA and protein level expression, in patients with recurrent pregnancy loss (RPL).
2. It is crucial to include an additional set of experiments using wild-type mice when analyzing the effects of the SIRT1 inhibitor EX527 on H4K16ac.
3. Does EX527 treatment partially restore placental development and improve reproductive outcomes in mice?
4. Similarly, when analyzing the effects of EX527 in htsos, it is necessary to include a control group treated with EX527 in the experimental setup.

Point-by-point response to Reviewer's Comments

We appreciate very much for your time and effort in handling this manuscript. We thank the reviewers for their constructive and thoughtful comments that helped strengthen the manuscript. In response to the Reviewer's concerns, we performed additional experiments and include substantial new data in the revised paper. In particular, we have established a new human trophoblast cell line with high CDX2 expression induced from hESC, and proved the applicability of the KAT8-H4K16ac-CDX2 axis in humans.

We believe that we have addressed all the reviewer's criticisms, and hope the revised manuscript is now acceptable for publication in *Nature Communications*. Changes in the text are highlighted in blue in the revision. In addition, Figure 6 and Supplementary Figure 5 are new.

Below, please find our detailed, point-by-point response to each of the Reviewer's comments.

REVIEWER COMMENTS

Reviewer #1 (Remarks to the Author):

General Comment:

The authors have elucidated the role of KAT-8 and H4K16ac in maintaining stem cell properties during placental development, using genetically modified mice, mouse TS cells, human placental tissues, and placental organoids. The mouse data is particularly conclusive, demonstrating the critical role of KAT8 in embryonic development through sophisticated experimental methods like gene modification and rescue experiments. The significance of these findings is noteworthy. The potential usefulness of EX527 treatment is also suggested, which could be of great importance. However, the human data seem less robust compared to the mouse data. If human data are to be included, improvement in this section is necessary.

Response: We greatly value the positive feedback and constructive suggestions provided by the reviewer regarding our work. In light of these suggestions, we have implemented necessary revisions to the draft. The responses to these revisions are detailed below.

Major Comments:

1. Recent RNA-seq studies suggest that CDX2 is not expressed in the first-trimester placenta (Vento-Tormo et al., Nature 2018; Suryawanshi et al., Science Adv 2018). The paper discusses relative expression levels using qPCR, but isn't the absolute expression of CDX2 quite low?

Response: Thank you for bringing this to our attention. We respectfully disagree with

the assertion that no CDX2 is expressed in the first-trimester placenta. After a thorough examination of the scRNAseq data referenced in the provided papers and an analysis of the scRNAseq data by Vento-Tormo *et al.*, our findings suggest otherwise. As shown in the graphs below, the presence of CDX2 was detected specifically within the villous cytotrophoblast (VCT) of placental samples, albeit at a relatively low expression level.

Upon closer scrutiny, we determined that the observed lower expression of CDX2 may be attributed to the timing of placental villi collection for scRNAseq profiling. These collections were conducted during the 6-11 weeks (Suryawanshi *et al.*, Science Adv 2018) or 6-14 weeks of gestation (Vento-Tormo *et al.*, Nature 2018). Notably, previous studies have documented that CDX2 expression gradually diminishes with trophoblast differentiation during placental development (Horii M *et al.*, PNAS. 2016; Paul S *et al.*, Placenta. 2017; Soncin F *et al.*, Development. 2018; Turco MY *et al.*, Development. 2019).

To further confirm the expression of CDX2 in first-trimester placenta, we performed additional immunofluorescence staining on placental villi collected at 6-7 weeks of pregnancy. The results, as shown in the images below, revealed CDX2 expression colocalized with the CTB marker TP63 at 6 weeks, with a noticeable decrease in expression by 7 weeks. This trend aligns with previously reported findings (Horii M *et*

al., PNAS, 2016; Soncin et al., Development, 2018), supporting the presence of CDX2 during early pregnancy (≤ 6 weeks), followed by a gradual decline in expression as gestational weeks progress and placental trophoblast differentiation occurs (>7 weeks).

Significantly, the organoids utilized in our study were collected specifically from placental villi at 6 weeks, faithfully reproducing the observed CDX2 expression pattern and percentage *in vivo* (refer to Supplemental Figure 5). Therefore, we are confident in the accuracy and feasibility of the qPCR analysis conducted on CDX2 levels within our organoid system. We have included these additional details in our revised manuscript (see Supplementary Fig. 6a, line 305-310).

2. While some papers report strong CDX2 staining in cytotrophoblast cells, most use the same monoclonal antibody (EPR2764Y) as in this study. It would be advisable to use other well-validated antibodies for detecting CDX2 in the human placenta.

Response: Thank you for your suggestion. As noted, the antibody utilized (EPR2764Y) in our work has demonstrated effectiveness across numerous studies, our CDX2 staining specifically reflect its expression pattern vary during pregnancy, with a decline observed as gestation progresses (see the above response and revised Supplementary Fig. 6a), underscoring the reliability of this antibody. However, in response to your suggestion, we purchased two additional CDX2 antibody (catalog number NB100-2136, ab101532) and conducted new IF experiments on human placental samples obtained at 6 weeks of gestation. As shown in the imagines below, both antibodies consistently labeled CDX2 expression in a comparable percentage and pattern to the one reported in our study, providing further support for the reliability and robustness of our results.

3. The proportion of CDX2 positive cells in Fig. 7J is quite low at 0.25%, raising questions about the accuracy and credibility of the expression.

Response: We appreciate your attention to detail. We apologize for the error in the labeling of the y-axis in the initial Fig. 7J (now revised Fig. 8j). It should have been labeled as 0, 10, 20, 30 to accurately indicate that the proportion of CDX2-positive cells is 25%, as depicted by the CDX2 staining in the initial Figure 7I (now revised Fig. 8i). We have made the correction to accurately reflect the data in the revised manuscript (see revised Figure 8j).

4. The paper mentions a KAT8-H4K16ac-CDX2 axis in humans, but there is no proof that CDX2 controls the proliferation of organoids. Is the expression of CDX2 too low or non-existent? For the human placental organoid section, comprehensive expression analysis similar to what was done for mice would be beneficial, particularly regarding KAT8 knockdown and RPL-TSO.

Response: We appreciate your thoughtful concerns. In our *Kat8^{fl/fl}; Elf5-Cre* mouse model, *Kat8* deletion was initiated early in trophoblast lineage formation concurrent with ELF5 expression (E4.5), leading to phenotypic manifestation post-embryo implantation primarily attributed to the loss of *Cdx2*. However, it's important to note

the disparities between our mouse model and human biology. In human studies, placental villi are typically obtained for organoid establishment between 6-8 weeks of pregnancy, a period where CDX2 expression is infrequent. While we believe that our observations on 6-week placental villi organoids are indicative of findings in mice, we acknowledge challenges in replicating all mouse TSC experiments regarding the KAT8-H4K16ac-CDX2 axis, due to sparse CDX2 expression in hTOs.

To overcome ethical limitations of collecting placental villi from <6-week pregnancies and validate KAT8-H4K16ac-CDX2 axis findings in human trophoblast cells, we induced a CDX2-high trophoblast cell line (iTSC) from hESCs, following a previously reported method (Mischler A et al. in J Biol Chem. 2021). As depicted in the newly generated figure below, CDX2 expression was confirmed in iTSC along with other TSC markers, TEAD4 and TFAP2C, indicating successful CDX2-high TSC model establishment. Using this cell line, we knocked down KAT8 expression via lentivirus-mediated shKAT8 infection. KAT8 knockdown notably decreased H4K16ac and CDX2 expression, as observed through western blot and IF staining. Additionally, Ki67 expression decreased, correlating with slower growth in shKAT8 cells compared to shNC cells.

Further experiments involved knockdown of KAT8, CDX2 overexpression, and EX527 addition in CDX2-high TSCs. Interestingly, decreased growth due to KAT8 knockdown was rescued by CDX2 induction. Furthermore, EX527 notably increased H4K16ac and CDX2 expression in KAT8 knockdown iTSCs. These experiments corroborated the presence of the KAT8-H4K16ac-CDX2 axis in human trophoblast cells, indicating its conservation. These additional experimental results have been incorporated into the revised manuscript (see revised Figure 6, lines 275-294).

Figure 6

Minor Comments:

1. Line 86: A reference is needed for "Extensive research has elucidated the pivotal role of H4K16ac in maintaining the pluripotency of embryonic stem cells (ESCs)."

Response: Thank you for your suggestion. We have added references to support the statement in the revised manuscript (see line 92).

2. Ref. 16: Indent confirmation is needed.

Response: Thank you for your careful reading. We have changed it in the revised manuscript (see line 694).

3. Line 108: More detail on the staining results in Fig. 1A would be beneficial, especially for understanding the localization of KAT8 at E9.5. Adding annotations in

the figure to indicate the magnified areas could improve clarity.

Response: Thank you for your kind suggestions. We have added annotations in the revised Figure 1a with magnified areas to improve clarity accordingly.

4. Line 117: The expression "loss of Kat8d/d" seems redundant.

Response: Thank you for pointing out his mistake, we have deleted d/d in the revised manuscript (see line 122).

5. Line 137: For Kat8d/d mice, "significantly reduced" is stated, but there is no statistical significant difference between the two types of mice.

Response: Thank you for the careful reading. We have included the figure legend of arrowheads in Figure 1G, which pointing out the missed H4K16ac and Ki67 staining in *Kat8^{d/d}* ExE cells. Accordingly, we have rewrote the sentence as follows in the revised manuscript: "Furthermore, the expression of Ki67, a marker of cellular proliferation, was undetectable in a subset of H4K16ac negative cells of Kat8d/d ExE at E5.5." (See lines 142-143)

6. Line 238: The increase in CDX2 does not appear to be dose-dependent.

Response: Thanks for pointing out this issue. We have modified the sentence reads as follows in the revised manuscript (see lines 241-242): "An evident elevation in H4K16ac and CDX2 levels was noted via western blot and IF upon treatment with Ex527 at 20 μ M or higher."

7. Line 294: A citation is needed.

Response: Thanks for your careful reading, we have included a ref in the revised manuscript (see line 332).

8. In Fig. 7K and Sup Fig. 5A, "Trophoblast stem cells" should be labeled as "Cytotrophoblast," as the organoids do not seem to be "Stem cell-derived" based on the methods in this study.

Response: Thank you for bringing it to our attention. The label has been adjusted accordingly, and we have replaced "trophoblast stem-like organoid (hTSO)" with "trophoblast organoid (hTO)" consistently throughout the revised manuscript.

9. Line 375: The statement "This decline in H4K16ac correlates negatively with SIRT1 expression" needs clarification. Is this observed in the current study? If based on

previous reports, a citation is required.

Response: Thank you for pointing out this error. We have removed this statement from the manuscript (see lines 414-415 in the revised manuscript).

Reviewer #2 (Remarks to the Author):

The study presented by the authors investigates the role of Kat8, a MYST family histone acetyltransferase, in regulating trophoblast self-renewal and differentiation during early gestation. The findings demonstrate that trophoblast-specific deletion of Kat8 results in extraembryonic ectoderm abnormalities and embryonic lethality. Through comprehensive analyses, including RNA-seq and CUT&Tag, the study uncovers KAT8's role in the transcriptional activation of the trophoblast stemness marker CDX2, mediated by acetylating H4K16. The rescue experiments with CDX2 overexpression and the use of the SIRT1 inhibitor, EX527, provide valuable insights into the regulatory mechanisms. Clinical correlation with recurrent pregnancy loss (RPL) and the amelioration of trophoblast organoid abnormalities with EX527 treatment further strengthen the significance of the study. However, additional experiments and thorough validation are recommended to strengthen the causal relationships proposed in the manuscript.

Response: We are encouraged by the positive comments of the reviewer on our work. Based on the suggestions, we made necessary changes to the draft and the responses to the revision were listed below.

1. The author should conduct research on the expression changes of CDX2, including mRNA and protein level expression, in patients with recurrent pregnancy loss (RPL).

Response: Thank you for your suggestion. As requested, we performed additional IF staining for CDX2 on chorionic villi tissues from RPL patients at approximately 6 weeks of gestation. Our results showed a notable decrease, with some samples showing undetectable levels of CDX2 expression in RPL chorionic villi compared to those from normal pregnancies (see revised Fig. 7f-g, lines 321-323).

2. It is crucial to include an additional set of experiments using wild-type mice when analyzing the effects of the SIRT1 inhibitor EX527 on H4K16ac.

Response: Thank you for your constructive suggestion. We have included the set of wild-type mice as indicated in the revised manuscript (see revised Supplementary figure 4a, lines 248-251).

3. Does EX527 treatment partially restore placental development and improve reproductive outcomes in mice?

Response: Thank you for your insightful questions. Our findings, as indicated in Figure 5 and supplementary Figure 5, demonstrate that EX527 application partially ameliorates trophoblast defects induced by *Kat8* knockout and extends extraembryonic ectoderm development from E5.5 to E8.5. To assess whether Ex527 improves reproductive outcomes in *Kat8^{kd/d}* mice, we examined late-stage embryo resorption and conducted genotype identification of offspring to ascertain the ultimate outcome. However, our results indicate that EX527 does not rescue late-stage embryonic lethality (see revised Supplementary Figs. 4g-h).

This finding is consistent with prior research indicating that elevated H4K16ac levels, as observed in *Sirt1* knockout models, disrupt trophoblast differentiation and result in embryonic death during mid-gestation (Kanaga Arul Nambi Rajan et al., Placenta 2018). It underscores the dynamic role of H4K16ac in placental development, wherein its equilibrium during trophoblast stemness and differentiation is finely regulated. Moreover, *Kat8* may also be crucial for TSC differentiation through mechanisms independent of H4K16ac, as discussed in the manuscript (see lines 389-391, and lines 414-415 in the Discussion part).

4. Similarly, when analyzing the effects of EX527 in htsos, it is necessary to include a control group treated with EX527 in the experimental setup.

Response: Thank you for your suggestion. We have included an EX527-treated control group in the hTO experiments. Our assessment of H4K16ac expression, CDX2 levels, and proliferation in this control group revealed that EX527 treatment did not exert a significant impact on organoid growth and organoid forming efficiency (see below images and graphs). These additional results have been included in the supplementary materials of the revised manuscript (see revised figure 5 and lines 348-352).

Reviewers' Comments:

Reviewer #1:

Remarks to the Author:

I have reviewed the revised manuscript and am satisfied with the changes.

The additional experiments and revisions have adequately addressed my initial concerns, making the manuscript now suitable for publication.

Reviewer #2:

Remarks to the Author:

All the questions I've been following have been answered and addressed. I don't have any other questions at the moment.